# ENCODINGS FOR PREDICTION-BASED NEURAL ARCHITECTURE SEARCH

## ABSTRACT

Predictor-based methods have substantially enhanced Neural Architecture Search (NAS) optimization, with the efficacy of these predictors largely influenced by the method of encoding neural network architectures. While traditional encodings used an adjacency matrix describing the graph structure of a neural network, novel encodings embrace a variety of approaches from unsupervised pretraining of latent representations to vectors of zero-cost proxies. In this paper, we categorize and investigate neural encodings from three main types: structural, learned, and score-based. Furthermore, we extend these encodings and introduce *unified encodings*, that extend NAS predictors to multiple search spaces. Our analysis draws from experiments conducted on over 1.5 million neural network architectures on NAS spaces such as NASBench-101 (NB101), NB201, NB301, Network Design Spaces (NDS), and TransNASBench-101. Building on our study, we present our predictor **FLAN**: **Fl**ow **A**ttention for **N**AS. FLAN integrates critical insights on predictor design, transfer learning, and *unified encodings* to enable more than an order of magnitude cost reduction for training NAS accuracy predictors. Our implementation and encodings for all neural networks are open-sourced at `https://anonymous.4open.science/r/flan_nas-433F/`.

## 1 INTRODUCTION

In recent years, Neural Architecture Search (NAS) has emerged as an important methodology to automate neural network design. NAS consists of three components: (1) a neural network search space that contains a large number of candidate Neural Networks (NNs), (2) a search algorithm that navigates that search space, and (3) optimization objectives such as NN accuracy and latency. A key challenge with NAS is its computational cost, which can be attributed to the sample efficiency of the NAS search algorithm, and the cost of evaluating each NN candidate. A vast array of search algorithms have been proposed to improve NAS sample efficiency, ranging from reinforcement learning (Zoph & Le, 2017), to evolutionary search (Pham et al., 2018), and differentiable methods (Liu et al., 2019). To reduce the evaluation cost of each NN candidate, prior work has utilized reduced-training accuracy (Zhou et al., 2020) zero-cost proxies (Abdelfattah et al., 2021), and accuracy predictors that are sometimes referred to as surrogate models (Zela et al., 2020). One of the most prevalent sample-based NAS algorithms utilizes accuracy predictors to both evaluate a candidate NN, and to navigate the search space. Recent work has clearly demonstrated the versatility and efficiency of *prediction-based NAS* (Dudziak et al., 2020; Lee et al., 2021), highlighting its importance. In this paper, we focus on understanding the makings of an efficient accuracy predictor for NAS, and we propose improvements that significantly enhance its sample efficiency.

An integral element within NAS is the encoding method to represent NN architectures. Consequently, an important question arises, *how can we encode neural networks to improve NAS efficiency?* This question has been studied in the past by (White et al., 2020), specifically investigating the effect of graph-based encodings such as adjacency matrices or path enumeration to represent NN architectures. However, recent research has introduced a plethora of new methods for encoding NNs which rely on concepts ranging from unsupervised auto-encoders, zero-cost proxies, and clustering NNs by computational similarity to learn latent representations. This motivates an updated study on NN encodings for NAS to compare their relative performance and to elucidate the properties of effecive encodings to improve NAS efficiency. We identify three key categories of encodings. **Structural** encodings (White et al., 2020) represent the graph structure of the NN architecture in the

Figure 1: The basic structure of an accuracy predictor highlights that many different types of encodings can be fed to the same prediction head to perform accuracy prediction.

form of an adjacency matrix or path enumeration, typically represented by an operation matrix to identify the operation at each edge or node. **Score-based** (Akhauri & Abdelfattah (2023)) encodings map architectures to a vector of *measurements* such as Zero-Cost Proxies (Lee et al.; Tanaka et al., 2020; Mellor et al., 2021). Finally, **Learned** encodings learn latent representations of the architecture space. They can be further bifurcated into ones that explicitly learn representations through large-scale unsupervised training (Yan et al., 2020; 2021) and ones that co-train neural encodings during the supervised training of an accuracy predictor (Ning et al., 2022; Guo et al., 2019). Figure 1 illustrates our taxonomy of encoding methods, and their role within a NAS accuracy predictor.

NN encodings are particularly important in the case of prediction-based NAS because they have a large impact on the effectiveness of training an accuracy predictor. For that reason, and due to the increasing importance of predictors within NAS, our work provides a comprehensive analysis of the impact of encodings on the sample-efficiency of NAS predictors. We validate our observations on 13 NAS design spaces, spanning 1.5 million neural network architectures across different tasks and data-sets. Furthermore, NAS predictors have the capability to *extend* beyond a single NAS search space through transfer learning (Akhauri & Abdelfattah, 2023), or more generally metalearning (Lee et al., 2021). In this regime, pretraining a NAS predictor on a readily-available search space (for example, a NAS benchmark), then transferring the predictor to a new search space could be achieved with very few samples on the new search space, resulting in even more efficient NAS. We posit that encodings play a major role in the efficacy of this approach and therefore, we empirically evaluate transfer predictors as part of our comprehensive study. **Our Contributions** are:

1. We categorize and study the performance of many different NN encoding methods in NAS accuracy prediction across 13 different NAS search spaces.
2. We propose a new hybrid encoder (called FLAN) that outperforms prior methods consistently on multiple NAS benchmarks. We demonstrate a $2.12\times$ improvement in NAS sample efficiency.
3. We create *unified* encodings that allow accuracy predictors to be transferred to new search spaces with a very low number of new accuracy samples. Notably, we are able to improve sample efficiency of predictor training by $46\times$ across three NAS spaces compared to trained-from-scratch predictors from prior work.
4. We generate and provide open access to structural, score-based, and learned encodings for over 1.5 million neural network architectures, spanning 13 distinct NAS spaces.

## 2 RELATED WORK

**Predictor-based NAS.** NAS consists of an evaluation strategy to fetch the accuracy of an architecture, and a search strategy to explore and evaluate novel architectures. Predictor-based NAS involves training an accuracy predictor which guides the architectural sampling using prediction scores of unseen architectures (Dudziak et al., 2020; White et al., 2021). Recent literature has focused on the sample efficiency of these predictors, with BONAS (Shi et al., 2020) using a GCN for accuracy prediction as a surrogate function of Bayesian Optimization, BRP-NAS (Dudziak et al. (2020)) employing a binary relation predictor and iterative sampling strategy. Recently, TA-GATES (Ning et al., 2022) employed learnable operation embeddings and introduced a method of updating embeddings akin to the training process of a NN to achieve state-of-the-art sample efficiency.

**NAS Benchmarks.** To facilitate NAS research, a number of NAS benchmarks have been released, both from industry and academia (Ying et al., 2019; Zela et al., 2020; Duan et al., 2021; Mehta et al.,

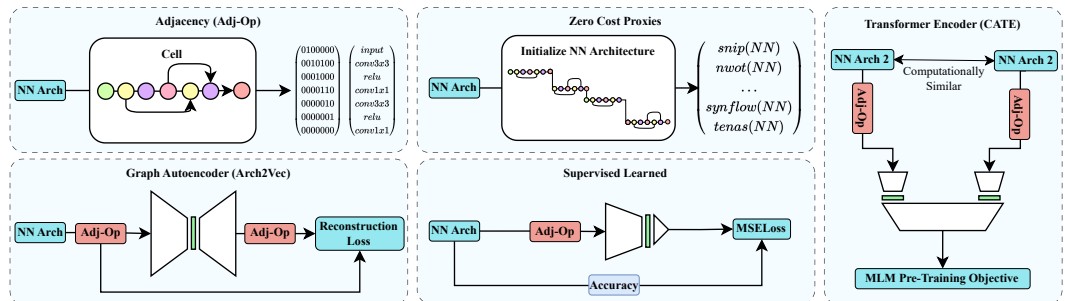

Figure 2: Illustration of important encoding methods that are discussed and evaluated in our work.

2022). These benchmarks contain a NAS search space and trained accuracy for each architecture on a specific task. Even though most of these benchmarks focus on cell-based search spaces for the image classification, they greatly vary in size (8k – 400k architectures) and NN connectivity. Additionally, more recent benchmarks have branched out to include other tasks (Mehrotra et al., 2021) and macro search spaces (Chau et al., 2022). Evaluations on a number of these benchmarks have become a standard methodology to test and validate NAS improvements without incurring the large compute cost of performing NAS on a new search space.

**NN Encodings.** There are several methods for encoding candidate NN architectures. Early NAS research focused on *structural encodings*, converting the adjacency and operation matrix representing the DAG for the candidate cell into a flattened vector to encode architectures (White et al., 2020). *Score-based methods* such as Multi-Predict (Akhauri & Abdelfattah, 2023) focus more on capturing broad architectural properties, by generating a vector consisting of zero-cost proxies and hardware-latencies to represent NNs. There have also been efforts in *unsupervised learned* encodings such as Arch2Vec (Yan et al., 2020), which leverages the graph auto-encoders to learn a compressed latent vector used for encoding a NN. Another method, CATE (Yan et al., 2021), leverages concepts from masked language modeling to learn latent encodings using computational-aware clustering of architectures using Transformers. Finally, many supervised accuracy predictors implicitly learn encodings as intermediate activations in the predictor. These *supervised learned* encodings have been generated most commonly with graph neural networks (GNNs) within accuracy predictors (Dudziak et al., 2020; Shi et al., 2020; Ning et al., 2022; Liu et al., 2022).

## 3 ENCODINGS

A basic NAS formulation aims to maximize an objective function $\ell : A \to \mathbb{R}$, where $\ell$ is a measure of NN accuracy for our purposes but can include performance metrics as well such as hardware latency (Dudziak et al., 2020). $A$ is a NN search space. During NAS, NN architectures $a \in A$ are *encoded* using some encoding function $e : A \to \mathbb{R}^d$, that represents a NN architecture as a d-dimensional tensor. While prior work (White et al., 2020) only considered a narrow definition of encodings wherein $e$ was a fixed transformation that was completely independent of $\ell$, we expand the definition to also consider encoding functions that are parameterized with $\theta$. This includes supervised training to minimize the empirical loss $\mathcal{L}$ on predicted values of $\ell$ to actual measurements: $\min_{\theta, \phi} \sum_{a \in A'} \mathcal{L}(F_\phi(e_\theta(a)), \ell(a))$, where $F : E \to \mathbb{R}$ is a prediction head that takes a learned encoding value $e(a)$ and outputs predicted accuracy $\ell'(a)$. Simply put, this allows us to evaluate part of an accuracy predictor as a form of encoding, for example, a learned graph neural network encoding function that is commonly used in predictor-based NAS (Ning et al., 2022). Our definition also includes the use of unsupervised training to learn a latent representation $r$, for example using an autoencoder which attempts to optimize $\min_{\theta, \phi} \sum_{a \in A'} \mathcal{L}(F_\theta^{enc}(e(a)), F_\phi^{dec}(r))$ using an encoder-decoder structure that is trained to recreate the graph-based structure of an NN (e.g. adjacency encoding) (Yan et al., 2020). Our broader definition of encodings allows us to compare many methods of NN encodings that belong to the four categorizations below—important encodings are illustrated in Figure 2.

**Structural** encodings capture the connectivity information of a NN exactly. White et al. (2020) investigate two primary paradigms for structural encodings, *Adjacency* and *Path* encodings. A neural network can have $n$ nodes, the adjacency matrix simply instantiates a $n \times n$ matrix, where each nodes connectivity with the other nodes are indicated. On the other hand, Path encodings represent

Figure 3: The FLAN predictor architecture showing dual graph flow mechanisms, independent updates of operation embeddings, and the capability to concatenate supplementary encodings.

an architecture based on the set of paths from input-to-output that are present within the architecture DAG. There are several forms of these encodings discussed further by White et al. (2020), including Path truncation to make it a fixed-length encoding. Their investigation reveals that Adjacency matrices are almost always superior at representing NNs.

**Score-based** encodings represent a neural network as a vector of measurements related to NN activations, gradients, or properties. These metrics were defined to be a vector of Zero-Cost Proxies (ZCPs) and hardware latencies (HWL) in MultiPredict (Akhauri & Abdelfattah, 2023), and used for accuracy and latency predictors respectively. Zero-cost proxies aim to find features of a NN that correlate highly with accuracy, whereas hardware latencies are fetched by benchmarking the architecture on a set of hardware platforms. Naturally, connectivity and choice of operations would have an impact on the final accuracy and latency of a model, therefore, these encodings implicitly capture architectural properties of a NN architecture, but contain no explicit structural information.

**Unsupervised Learned** encodings are representations that aim to distil the structural properties of a neural architecture to a latent space without utilizing accuracy. Arch2Vec (Yan et al., 2020) introduces a variational graph isomorphism autoencoder to learn to regenerate the adjacency and operation matrix. CATE (Yan et al., 2021) is a transformer based architecture that uses computationally similar architecture pairs (FLOPs or parameter count) to learn encodings. With two computationally similar architectures, a transformer is tasked to predict masked operations for the pairs, which skews these encodings to be similar for NNs with similar computational complexity. Unsupervised Learned encodings are typically trained on a large number of NNs because NN accuracy is not used.

**Supervised Learned** encodings refer to representations that are implicitly learned in a supervised fashion as a predictor is trained to estimate accuracy of NN architectures. These encodings are representations that evolve and continually adapt as more architecture-accuracy pairs are used to train an accuracy predictor. Supervised Learned encodings are more likely to exhibit a high degree of bias towards the specific task on which they are trained, potentially limiting their generality when extending/transferring a predictor to a different search space.

**Unified Encodings.** Multi-Predict (Akhauri & Abdelfattah, 2023) and GENNAPE (Mills et al., 2022) introduce encodings that can represent arbitrary NNs across multiple search spaces. Further, CDPLiu et al. (2022) introduces a predictor trained on existing NAS benchmark data-sets, and is then used to find architectures in large-scale search spaces. In this paper, we look at unified encoding methods that can work across cell-based search spaces. An approach to unified cell-based NAS can enable knowledge reuse across several NAS spaces, and enable prototyping of novel search spaces. To make our encodings unified, we append unique numerical indices to the cell-based encoding of each search space. This *unified operation space* can enable training any of our studied encodings across multiple search spaces.

## 4 FLAN: FLOW ATTENTION NETWORKS FOR NAS

Our empirical evaluation (yet to be presented in Table 2) shows that Supervised Learned encoders often out-perform other encoding methods. This is somewhat expected because they have access to the accuracies of NNs in the search space. However, training candidate NN architectures can be fairly expensive, and it is not always feasible to obtain accuracies of a sufficient number of NNs, therefore, we focus on the sample efficiency of the accuracy predictors. In this section, we introduce **FLAN**: a hybrid encoding architecture which draws on our empirical analysis to deliver state-of-

the-art sample efficiency for accuracy prediction. We carefully tune the predictor architecture so that it can be used reliably as a vehicle to investigate and compare existing and new hybrid encoding schemes as well as unified encodings. Figure 3 shows the FLAN architecture, described further in this section. FLAN combines successful ideas from prior graph-based encoders (Dudziak et al., 2020; Ning et al., 2022) and further improves upon them through dual graph-flow mechanisms. In addition to learning an implicit NN encoding, FLAN can be supplemented with additional encodings arbitrarily through concatenation before the predictor head as shown in Figure 3.

## 4.1 GNN ARCHITECTURE

Compared to Multi-Layer Perceptrons (MLPs), Graph Convolutional Networks (GCN) improve prediction performance, as shown in Table 2. We employ an architectural adaptation inspired by (Ming Chen et al., 2020), referred to as 'Dense Graph Flow' (DGF) (Ning et al., 2023). Empirical analysis, detailed in Table 16, reveals that substantial enhancements in predictor performance can be realized through the integration of residual connections (Kipf & Welling, 2017) within DGF. Further, we add another node propagation mechanism based on graph attention to facilitate inter-node interaction. Empirical results in Table 1 shows that the combination of both graph flows typically yields the best results.

**Dense Graph Flow (DGF):** DGF employs residual connections to counteract over-smoothing in GCNs, thereby preserving more discriminative, localized information. Formally, given the input feature matrix for layer $l$ as $X^l$, the adjacency matrix $A$, and the operator embedding $O$, with parameter and bias as $W_o^l$, $W_f^l$, and $b_f^l$ respectively, the input feature matrix for the $(l+1)^{th}$ layer is computed as follows ($\sigma$ is the sigmoid activation):

$$X^{l+1} = \sigma(OW_o^l) \odot (A(X^l W_f^l)) + (X^l W_f^l) + b_f^l \tag{1}$$

**Graph Attention (GAT):** Unlike DGF, which employs a linear transform $W_o^l$ to apply learned attention to the operation features, GAT (Veličković et al., 2018) evaluates pairwise interactions between nodes through an attention layer during information aggregation. The input to the $l^{th}$ layer is a set of node features (input feature matrix) $X^l$, to transform the input to higher level-features, a linear transform paramterized by the projection matrix $W_p^l$ is applied to the nodes. This is followed by computing the self-attention for the node features with a shared attentional mechanism $a$. The output $X^{l+1}$ is thus calculated as follows:

$$\text{Attn}_j(X^l) = \text{softmax}(\text{LeakyReLU}(A_j \cdot a(W_p^l X^l \cdot W_p X_j^l))) \cdot W_p X_j^l \tag{2}$$

$$X^{l+1} = \text{LayerNorm}\left(\sigma(OW_o^l) \odot \sum_{j=1}^{n} \text{Attn}_j(X^l)\right) \tag{3}$$

where $\text{Attn}_j$ are the normalized attention coefficients, $\sigma$ denotes the sigmoid activation function. To optimize the performance of GATs, we incorporate the learned operation attention mechanism $W_o$ from Equation 1 with the pairwise attention to modulate the aggregated information and LayerNorm to improve stability during training.

The primary components of FLAN which significantly boost predictor performance are the DGF residual connection, the learned operation attention mechanism $W_o$ in DGF and GAT and the pairwise attention in the GAT module. These modules are ensembled in the overall network architecture, and repeated 5 times. Additionally, the NASBench-301 and Network Design Spaces (NDS) search spaces (Radosavovic et al., 2019) provide benchmarks on large search spaces of two cell architectures, the normal and reduce cells. We train predictors on these search spaces by keeping separate DGF-GAT modules for the normal and reduce cells, and adding the aggregated outputs.

| Forward | Backward | NB101 | NB201 | NB301 | Amoeba | PNAS | NASNet | DARTS$_{FixWD}$ | ENAS$_{FixWD}$ | TB101 |
|---------|----------|-------|-------|-------|--------|------|--------|-----------------|----------------|-------|
| DGF | DGF | 0.708 | 0.798 | 0.712 | 0.420 | 0.375 | **0.419** | 0.463 | 0.479 | **0.793** |
| GAT | GAT | 0.653 | 0.772 | 0.793 | 0.375 | 0.357 | 0.313 | 0.544 | 0.459 | 0.745 |
| DGF+GAT | DGF | 0.718 | 0.0.806 | 0.811 | 0.385 | 0.300 | 0.317 | 0.552 | 0.525 | 0.764 |
| **DGF+GAT** | **DGF+GAT** | **0.732** | **0.820** | **0.820** | **0.459** | **0.422** | 0.387 | **0.557** | **0.568** | 0.754 |

Table 1: Using both node propagation methods within FLAN works best. Table shows Kendall Tau coeff. of accuracy predictors trained on 128 NNs and tested on the remainder of each search space.

| Classification | Encoder | NB101 (Portion of 7290 samples) | | | NB201 (Portion of 7813 samples) | | | NB301 (Portion of 5896 samples) | | | ENAS (Portion of 500 samples) | | |
|---|---|---|---|---|---|---|---|---|---|---|---|---|---|
| | | 1% | 5% | 10% | 0.1% | 0.5% | 1% | 0.5% | 1% | 5% | 5% | 10% | 25% |
| Structural | ADJ | 0.327 | 0.464 | 0.514 | 0.047 | 0.273 | 0.382 | 0.275 | 0.401 | 0.537 | 0.057 | 0.060 | 0.089 |
| | Path | 0.387 | 0.696 | 0.752 | 0.133 | 0.307 | 0.396 | - | - | - | - | - | - |
| Score | ZCP | 0.591 | 0.662 | 0.684 | 0.248 | 0.397 | 0.376 | 0.286 | 0.272 | 0.367 | 0.387 | 0.458 | 0.540 |
| Unsupervised Learned | Arch2Vec | 0.210 | 0.346 | 0.345 | 0.046 | 0.165 | 0.144 | 0.174 | 0.228 | 0.379 | 0.202 | 0.228 | 0.324 |
| | CATE | 0.362 | 0.458 | 0.467 | 0.462 | 0.551 | 0.571 | 0.388 | 0.349 | 0.417 | 0.200 | 0.279 | 0.410 |
| Supervised Learned | GCN | 0.366 | 0.597 | 0.692 | 0.246 | 0.311 | 0.408 | 0.095 | 0.128 | 0.267 | 0.230 | 0.314 | 0.428 |
| | GATES | 0.632 | 0.749 | 0.769 | 0.430 | 0.670 | 0.757 | 0.561 | 0.606 | 0.691 | 0.340 | 0.428 | 0.527 |
| | FLAN | 0.685 | 0.776 | 0.782 | 0.491 | 0.713 | 0.781 | 0.539 | 0.537 | 0.698 | 0.182 | 0.319 | 0.474 |
| Hybrid | TAGATES | 0.668 | 0.774 | 0.783 | 0.538 | 0.670 | 0.773 | 0.572 | 0.635 | 0.712 | 0.345 | 0.440 | 0.548 |
| | $\text{FLAN}_{ZCP}$ | 0.700 | 0.788 | 0.786 | 0.535 | 0.716 | 0.773 | 0.573 | 0.656 | 0.721 | 0.402 | 0.504 | 0.578 |
| | $\text{FLAN}_{Arch2Vec}$ | 0.653 | 0.752 | 0.762 | 0.501 | 0.706 | 0.772 | 0.417 | 0.509 | 0.688 | 0.138 | 0.163 | 0.397 |
| | $\text{FLAN}_{CATE}$ | 0.681 | 0.783 | 0.778 | 0.485 | 0.709 | 0.783 | 0.527 | 0.502 | 0.702 | 0.188 | 0.300 | 0.479 |
| | $\text{FLAN}_{CAZ}$ | 0.677 | 0.780 | 0.782 | 0.503 | 0.706 | 0.786 | 0.517 | 0.537 | 0.698 | 0.354 | 0.457 | 0.542 |

Table 2: A comparative study of accuracy predictors when utilizing different encoding methods. Table shows Kendall Tau correlation coefficient of predictors. $\text{FLAN}_X$ refers to the FLAN encoder with supplemental X encodings. 9 trials for FLAN experiments.

## 4.2 OPERATION EMBEDDINGS

In a NN architecture, each node or edge can be an operation such as convolution, maxpool. GNNs generally identify these operations with a one-hot vector as an attribute. However, different operations have widely different characteristics. To model this, TA-GATES (Ning et al., 2022) operation embedding tables that can be updated independently from predictor training. Figure 3 depicts the concept of an iterative operation embedding update in more detail. Before producing an accuracy prediction, there are $T$ time-steps (iterations) in which the operation embeddings are updated and refined. In each iteration, the output of GNN flow is passed to a 'Backward GNN Flow' module, which performs a backward pass using a transposed adjacency matrix. The output of this backward pass, along with the encoding is provided to a learnable transform that provides an update to the operation embedding table. This iterative refinement, conducted over specified time steps, ensures that the encodings capture more information about diverse operations within the network. Refer to A.8 for a detailed ablation study focusing on vital aspects of the network design.

## 4.3 FLAN ENCODINGS

**Supplemental Encodings:** Supervised learned encodings are representations formed by accessing accuracies of NN architectures. While structural, score-based and unsupervised learned encodings do not carry information about accuracy, they can still be used to distinguish between NN architectures. For instance, CATE (Yan et al., 2021) learns latent representations by computational clustering, thus providing CATE may contextualize the computational characteristics of the architecture. ZCP provides architectural-level information by serving as proxies for accuracy. Consequently, supplemental encodings can optionally be fed into the MLP prediction head after the node aggregation as shown in Figure 3. We find that using architecture-level ZCPs can significantly improve the sample-efficiency of predictors. CAZ refers to the encoding resulting from the concatenation of CATE, Arch2Vec and ZCP.

**Unified Encodings and Transferring Predictors:** Transferring knowledge between different search spaces can enhance the sample efficiency of predictors. However, achieving this is challenging due to the unique operations and macro structures inherent to each search space. To facilitate cross-search-space prediction, a unified operation space is crucial. Our methodology is straightforward; we concatenate a unique search space index to *each* operation, creating distinctive operation vectors. These vectors can either be directly utilized by predictors as operation embeddings or be uniquely indexed by the operation embedding table. Note that the training time for FLAN is less than 10 minutes on a single GPU, it is thus straightforward to regenerate an indexing that supports more spaces, and re-train the predictor. It is noteworthy that ZCPs *inherently* function as unified encodings by measuring broad architectural properties of a neural network (NN). Conversely, Arch2Vec and CATE are cell-based encoders. To accommodate this, we developed new encodings for Arch2Vec and CATE within a combined search space of 1.5 million NN architectures from all our NAS benchmarks. We provide predictor training and NAS results for all spaces in Sections A.9 & A.6, and provide a sub-set of these results in the experiments section to compare easily to related work. To

| Samples | FLAN | + Arch2Vec | + CATE | + ZCP | + CAZ | FLAN | + Arch2Vec | + CATE | + ZCP | + CAZ |
|---|---|---|---|---|---|---|---|---|---|---|
| | | | NASBench-101 | | | | | NASBench-201 | | |
| 8 | 0.381 | 0.274 | 0.441 | 0.459 | **0.479** | 0.410 | 0.419 | 0.427 | 0.429 | **0.429** |
| 16 | 0.507 | 0.445 | 0.407 | **0.550** | 0.468 | 0.612 | 0.616 | 0.610 | 0.612 | **0.618** |
| 32 | 0.572 | 0.512 | 0.575 | **0.602** | 0.590 | 0.691 | 0.692 | 0.691 | **0.694** | 0.688 |
| 64 | 0.635 | 0.606 | 0.657 | **0.694** | 0.677 | 0.760 | 0.765 | 0.769 | **0.769** | 0.757 |
| 128 | 0.717 | 0.691 | 0.720 | **0.758** | 0.740 | 0.826 | 0.823 | 0.820 | 0.819 | **0.827** |
| | | | DARTS | | | | | DARTS$_{FixWD}$ | | |
| 8 | 0.027 | 0.046 | 0.044 | **0.158** | 0.101 | 0.100 | 0.118 | 0.119 | **0.121** | 0.113 |
| 16 | 0.073 | 0.090 | 0.085 | 0.289 | **0.299** | 0.172 | 0.152 | 0.169 | **0.175** | 0.167 |
| 32 | 0.192 | 0.164 | 0.182 | 0.376 | **0.389** | 0.321 | 0.232 | 0.320 | **0.392** | 0.256 |
| 64 | 0.389 | 0.339 | 0.425 | **0.546** | 0.517 | 0.475 | 0.385 | 0.483 | **0.479** | 0.453 |
| 128 | 0.487 | 0.400 | 0.531 | 0.584 | **0.558** | 0.554 | 0.456 | 0.483 | **0.556** | 0.544 |
| | | | ENAS | | | | | ENAS$_{FixWD}$ | | |
| 8 | 0.044 | 0.043 | 0.045 | 0.093 | **0.098** | 0.165 | 0.114 | 0.140 | 0.161 | **0.176** |
| 16 | 0.145 | 0.066 | 0.146 | **0.335** | 0.271 | 0.267 | 0.217 | 0.310 | **0.350** | 0.335 |
| 32 | 0.264 | 0.138 | 0.264 | **0.434** | 0.380 | 0.387 | 0.244 | 0.403 | **0.415** | 0.392 |
| 64 | 0.342 | 0.291 | 0.373 | **0.514** | 0.479 | 0.470 | 0.364 | 0.502 | **0.504** | 0.461 |
| 128 | 0.458 | 0.385 | 0.486 | **0.568** | 0.538 | 0.528 | 0.465 | 0.542 | **0.563** | 0.533 |

Table 3: A study highlighting the benefit of providing supplementary encodings in improving sample efficiency of FLAN. Table shows Kendall Tau correlation coefficient averaged over 9 trials.

realize such a transfer of predictors from one search space to another, a predictor, initially trained on the source search space, is adapted using the unified operation encodings and subsequently retrained on the target design space. This is denoted by a T superscript in our experiments section.

## 5 EXPERIMENTS

We investigate the efficacy of encodings on neural networks on 13 search spaces, including NB101, NB201, NB301, 9 search spaces from NDS and TransNASBench-101 Micro. All of our experiments follow the Best Practices for NAS checklist (Lindauer & Hutter, 2019), detailed in Appendix A.1. Contrary to prior work, we generate encodings for all architectures in the search space for evaluation. To effectively evaluate encodings on these spaces, we generate and open-source the CATE, Arch2Vec and Adjacency representations for 1487731 NN architectures. NAS-Bench-Suite-Zero (Krishnakumar et al., 2022a) introduces a data-set of 13 zero cost proxies across 28 tasks, totalling 44798 architectures. We supplement those with Zero-Cost Proxies for 487731 NN architectures to facilitate our experiments. Building on previous studies, we adopt Kendall's Tau (KDT) rank correlation coefficient relative to ground-truth accuracy as the primary measure of predictive ability. We use a pair-wise hinge ranking loss to train our predictors (Ning et al., 2022). We use these encodings and input them to a 3-layer MLP prediction head with ReLU nonlinearity except for the output layer.

**Encoding Study:** Table 2 broadly evaluates the different categories of encoders. To compare existing literature consistently, we follow the same experimental set-up as TA-GATES (Ning et al., 2022), using a sub-set of each search space. For each search space, like NB101, we train encoders on a fraction of the data, such as 1% of 7290 (72 architectures), and then test on all 7290 test sample architectures for NB101. This method applies the same number of training and test samples for each space. Our results show that Supervised Learned encodings perform best, especially when supplemented with additional encodings with our Hybrid encodings. rbtlrom Figure 9, we can see that score-based encodings typically help prediction with low sample count but there are diminishing returns with more training samples. Our best encoding, FLAN$_{ZCP}$, delivers up-to a 15% improvement in Kendall-Tau correlation compared to the best previous result from TA-GATES. The results highlight the efficacy of Supervised Learned encodings, and the importance of GNN enhancements such as residual connections and the dual graph flow mechanisms introduced in FLAN.

**Supplemental Encodings:** A key reason for developing FLAN is to test combinations of different encodings. In Table 3, we look at the impact of supplementing our baseline predictor with Arch2Vec, CATE and ZCP on 6 different NAS search spaces. Note that 'CAZ' means we concatenate all three encodings. This setting is more challenging than Table 2 because we report test accuracy on the entire NAS search space instead of limiting our tests to ∼500–7000 NNs. We also increase the number

| Samples | FLAN$^T$ | + Arch2Vec | + CATE | + ZCP | + CAZ | FLAN$^T$ | + Arch2Vec | + CATE | + ZCP | + CAZ |
|---|---|---|---|---|---|---|---|---|---|---|
| | NASBench-201 $\longrightarrow$ **NASBench-101** | | | | | NASBench-101 $\longrightarrow$ **NASBench-201** | | | | |
| 2 | 0.525 | 0.298 | **0.532** | 0.470 | 0.443 | **0.643** | 0.483 | 0.488 | 0.399 | 0.590 |
| 4 | 0.511 | 0.406 | **0.517** | 0.459 | 0.423 | **0.620** | 0.523 | 0.490 | 0.436 | 0.529 |
| 8 | **0.536** | 0.471 | 0.512 | 0.517 | 0.486 | **0.647** | 0.567 | 0.606 | 0.519 | 0.605 |
| 16 | **0.580** | 0.523 | 0.562 | 0.542 | 0.534 | 0.692 | 0.645 | 0.649 | 0.640 | **0.716** |
| | ENAS $\longrightarrow$ **DARTS** | | | | | PNAS$_{FixWD}$ $\longrightarrow$ **DARTS$_{FixWD}$** | | | | |
| 2 | 0.494 | 0.514 | 0.505 | **0.581** | 0.546 | 0.418 | 0.294 | 0.415 | **0.454** | 0.355 |
| 4 | 0.503 | 0.440 | 0.478 | **0.543** | 0.513 | 0.394 | 0.312 | 0.379 | **0.440** | 0.419 |
| 8 | 0.506 | 0.521 | 0.525 | **0.585** | 0.533 | 0.372 | 0.353 | 0.392 | **0.432** | 0.385 |
| 16 | 0.515 | 0.483 | 0.484 | **0.594** | 0.552 | 0.365 | 0.369 | 0.389 | **0.436** | 0.416 |
| | DARTS $\longrightarrow$ **ENAS** | | | | | NASNet $\longrightarrow$ **ENAS$_{FixWD}$** | | | | |
| 2 | 0.452 | 0.479 | 0.444 | **0.587** | 0.439 | 0.316 | 0.247 | 0.435 | **0.456** | 0.421 |
| 4 | 0.398 | 0.359 | 0.391 | **0.497** | 0.374 | 0.258 | 0.300 | 0.386 | **0.447** | 0.367 |
| 8 | 0.390 | 0.431 | 0.409 | **0.530** | 0.418 | 0.252 | 0.398 | 0.380 | **0.460** | 0.404 |
| 16 | 0.432 | 0.403 | 0.433 | **0.533** | 0.368 | 0.340 | 0.412 | 0.389 | **0.436** | 0.422 |

Table 4: A study demonstrating the effectiveness of transferring predictors from one search-space to another (Source $\longrightarrow$ Target) using our Unified encodings. Table shows Kendall Tau correlation averaged over 9 trials. In this paper, 512 samples are used from the source space.

| | From Scratch | Transfer From TB101 Class Scene | | | | | From Scratch | Transfer From NDS CIFAR-10 | | | |
|---|---|---|---|---|---|---|---|---|---|---|---|
| Samples | 16 | 4 | 6 | 8 | 16 | Samples | 16 | 2 | 4 | 8 | 16 |
| AutoEncoder | 0.456 | 0.758 | **0.788** | 0.767 | 0.767 | Amoeba | 0.053 | **0.633** | 0.581 | 0.574 | 0.580 |
| Class Object | 0.404 | **0.807** | 0.776 | 0.788 | 0.775 | DARTS | 0.121 | 0.326 | 0.332 | 0.372 | **0.464** |
| Jigsaw | 0.350 | 0.757 | 0.760 | **0.773** | 0.736 | ENAS | 0.171 | 0.519 | 0.458 | 0.497 | **0.545** |
| Room Layout | 0.391 | **0.786** | 0.782 | 0.770 | 0.770 | NASNet | 0.103 | 0.349 | 0.362 | 0.376 | **0.471** |
| Segment Semantic | 0.644 | **0.794** | 0.788 | 0.786 | 0.776 | PNAS | 0.076 | 0.343 | 0.350 | 0.416 | **0.488** |

Table 5: A study demonstrating the effectiveness of predictor transfer across tasks within TB101 Micro and from NDS CIFAR10 to ImageNet. Average Kendall Tau correlation over 9 trials.

of NAS spaces to 6 in this experiment, with more comprehensive results on our remaining datasets available in the appendix. Supplementing FLAN with ZCP encoding has a significant impact on ranking quality. For the NDS search spaces (NASNet, AmoebaNet, PNAS, ENAS, DARTS), there is a macro-level network depth $d$ and initial filter width $w$ hyper-parameter that affect model complexity. We incorporate these as a normalized w-d vector to the predictor after the aggregation node, similar to adding supplemental encodings. This significantly improves ranking quality on the NDS search spaces, and indicates that providing macro-level context within the network definition helps prediction. We use this observation in our next experiment when creating Unified Encodings. Over baseline FLAN, incorporating ZCP encoding improves predictor accuracy by 47% on average.

**Cross-Search Space Transfer:** As evident in Table 3, the ranking quality for several design spaces can be really low when training from scratch with very few samples. This is likely because 8 samples are not sufficiently representative to train a generalizable predictor. We therefore study FLAN$^T$, which utilizes Unified encodings as discussed in Section 4.3. ZCP is already *unified*, as it is applied on a full NN not just to one cell. From Table 4, we can see that pre-training FLAN on a source space with unified encodings can enable a significant improvement in accuracy, especially at very low sample counts. In this table, we report results for a range of source spaces to target spaces. In the Appendix, we discuss why NASBench-201 $\longrightarrow$ NASBench-101 transfer (and vice-versa) does not benefit from these encodings. We also provide results of transferring to target spaces from a single source space. For example, training on ENAS and transferring to DARTS can result in a Kendall Tau of 0.58 with only 1-2 samples, as opposed to a KDT of 0.58 with 128 samples when training from scratch, a $128\times$ improvement in sample efficiency. This trend holds across search spaces, as indicated in Table 10. To investigate whether this cross-search space transfer is effective across tasks, Table 5 presents the effectiveness of transferring from Class Scene to 5 other tasks on the TransNASBench-101 Micro benchmark. In this, we also present the effectiveness of transferring predictors from NDS CIFAR-10 to ImageNet all of which outperforms training from scratch with 16 samples. To compare with prior work, we also conduct the same study with the sample-sizes described in TA-GATES. Figure 8 looks at NB-101, NB-201 and ENAS spaces. On this setting,

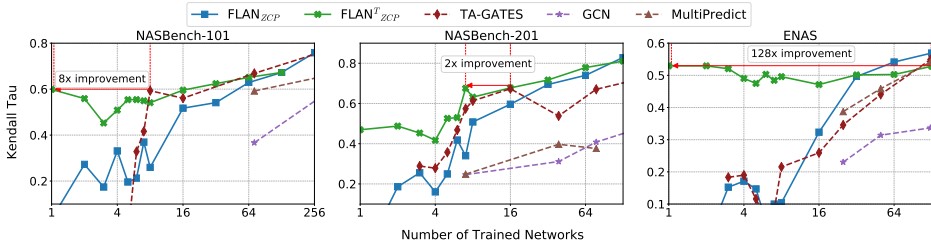

Figure 4: Prediction accuracy with different numbers of trained NNs. We investigate the impact of supplemental and unified encodings with FLAN, and compare to prior work. X-axis is logarithmic. Source search-spaces are the same as in Table 4.

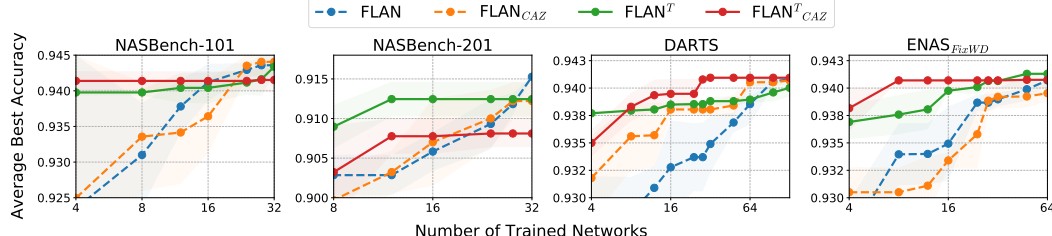

Figure 5: End-to-end NAS with different predictors using an iterative sampling algorithm. FLAN$^T$ improves performance in low sample count region. Source search-spaces are the same as Table 4.

we improve sample efficiency by $46\times$ on average across the three NAS spaces presented in the figure. Note that our improvements do not include the cost to pre-train the predictor on the source space intentionally because this can be considered a one-time cost and can be performed on a NAS benchmark instead of a real search space.

**Neural Architecture Search:** To gauge the sample-efficiency of FLAN in practice, we implement NAS search using the iterative sampling algorithm introduced by Dudziak et al. (2020). With a budget of $n$ models per iteration and $m$ models in the search space, we use our predictor to rank the entire search space, and then select the best $\frac{n}{2}$ models. To have a fair exploration-exploitation trade-off, we sample the next $\frac{n}{2}$ models from the top $max(512, \frac{m}{2^i})$ models, where $i$ is the iteration counter. Table 6 compares our results to the best sample-based NAS results found in the literature. We achieve the same test accuracy as Zero-Cost NAS (W) - Rand (3k) Abdelfattah et al. (2021) with $2.12\times$ fewer samples on end-to-end NAS with FLAN$_{CAZ}$. Further, we compare the NAS efficiency of different encoding methods within our framework in Figure 9. We find that transfer learning helps in general, with supplemental encodings (FLAN$_{CAZ}^T$) providing the best average performance.

| | BONAS | Aging Evo. (AE) | BRPNAS | Zero-Cost NAS (W) | | FLAN$_{CAZ}$ | | | | FLAN$_{CAZ}^T$ | | | |
| | | | | AE (15k) | RAND (3k) | | | | | | | | |
|---|---|---|---|---|---|---|---|---|---|---|---|---|---|
| Trained models | 1000 | 418 | 140 | 50 | 34 | 2 | 8 | 16 | 50 | 2 | 8 | 19 | 50 |
| Test Acc. [%] | 94.22 | 94.22 | 94.22 | 94.22 | 94.22 | 91.08 | 93.58 | 94.22 | 94.84 | 94.16 | 94.16 | 94.22 | 94.34 |

Table 6: A study on the number of trained models required to achieve a specified Test Accuracy.

## 6 CONCLUSION

We presented a comprehensive study of NN encoding methods, demonstrating their importance in enhancing the ranking quality and sample efficiency of NAS accuracy predictors. We found that Supervised Learned encodings that were co-trained with an accuracy predictor performed best, when compared to Structural, Score-based, or Unsupervised Learned encodings. This motivated our design of FLAN: a GNN-based NN encoding architecture with dual node propagation mechanisms and residual connections—empirically shown to outperform prior methods across different NAS benchmarks. We used FLAN to test NAS predictors in two new settings. First, we showed that *supplementary encodings* could be combined with FLAN to enhance its accuracy by up to 47%. Score-based supplemental encodings (ZCP) helped most in this setting. We additionally used FLAN to transfer accuracy predictors from one search space to another, demonstrating a $46\times$ average improvement in sample efficiency, and a $2.12\times$ improvement in practical NAS sample efficiency.

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

# A APPENDIX

## A.1 BEST PRACTICES FOR NAS

(White et al., 2020; Li & Talwalkar, 2019; Ying et al., 2019; Yang et al., 2020) discuss improving reproducibility and fairness in experimental comparisons for NAS. We thus address the sections released in the NAS best practices checklist by (Lindauer & Hutter, 2019).

- **Best Practice: Release Code for the Training Pipeline(s) you use:** We release code for our Predictor, CATE, Arch2Vec encoder training set-up.

- **Best Practice: Release Code for Your NAS Method:** We release our code publicly for the BRP-NAS style NAS search. We do not introduce a new NAS method.

- **Best Practice: Use the Same NAS Benchmarks, not Just the Same Datasets:** We use the NASBench-101, NASBench-201, NASBench-301, NDS and TransNASBench-101 datasets for evaluation. We also use a sub-set of Zero Cost Proxies from NAS-Bench-Suite-Zero.

- **Best Practice: Run Ablation Studies:** We run ablation studies for the design of FLAN in Table 17, Table 21, Table 15 and Table 18. We conduct ablation studies with different supplementary encodings in the main paper.

- **Best Practice: Use the Same Evaluation Protocol for the Methods Being Compared:** We use the same evaluation protocol as TAGATES when comparing encoders across literature. We provide additional larger studies that all follow the same evaluation protocol.

- **Best Practice: Evaluate Performance as a Function of Compute Resources:** In this paper, we study the sample efficiency of encodings. We report results in terms of the 'number of trained models required'. This directly correlates with compute resources, depending on the NAS space training procedure.

- **Best Practice: Compare Against Random Sampling and Random Search:** We propose a predictor - encoder design methodology, not a NAS method. We use SoTA BRP-NAS style NAS algorithm for comparing with existing literature.

- **Best Practice: Perform Multiple Runs with Different Seeds:** Our appendix contains information on number of trials as well as reproduction of tables in the main paper with standard deviation.

- **Best Practice: Use Tabular or Surrogate Benchmarks If Possible:** All our evaluations are done on publicly available Tabular and Surrogate benchmarks.

## A.2 NEURAL ARCHITECTURE DESIGN SPACES

In this paper, multiple distinct neural architecture design spaces are studied. Both NASBench-101(Ying et al., 2019) and NASBench-201(Dong & Yang, 2020) are search spaces based on cells, comprising 423,624 and 15,625 architectures respectively. NASBench-101 undergoes training on

| Search space | Tasks | Num. ZC proxies | Num. architectures | Total ZC proxy evaluations |
|---|---|---|---|---|
| NAS-Bench-101 | 1 | 13 | 423 625 | 5 507 125 |
| NAS-Bench-201 | 1 | 13 | 15 625 | 203 125 |
| DARTS | 1 | 13 | 5000 | 65 000 |
| ENAS | 1 | 13 | 4999 | 64 987 |
| PNAS | 1 | 13 | 4999 | 64 987 |
| NASNet | 1 | 13 | 4846 | 62 998 |
| AmoebaNet | 1 | 13 | 4983 | 64 779 |
| DARTS$_{FixWD}$ | 1 | 13 | 5000 | 65 000 |
| DARTS$_{LRWD}$ | 1 | 13 | 5000 | 65 000 |
| ENAS$_{FixWD}$ | 1 | 13 | 5000 | 65 000 |
| PNAS$_{FixWD}$ | 1 | 13 | 4559 | 59 267 |
| TransNASBench-101 Micro | 7 | 12 | 4096 | 344 064 |
| **Total** | 18 | 13 | 512 308 | 6 631 332 |

Table 7: Overview of ZC proxy evaluations in our work. ZCP for TransNASBench-101 Micro and NASBench201 are borrowed from Krishnakumar et al. (2022b).

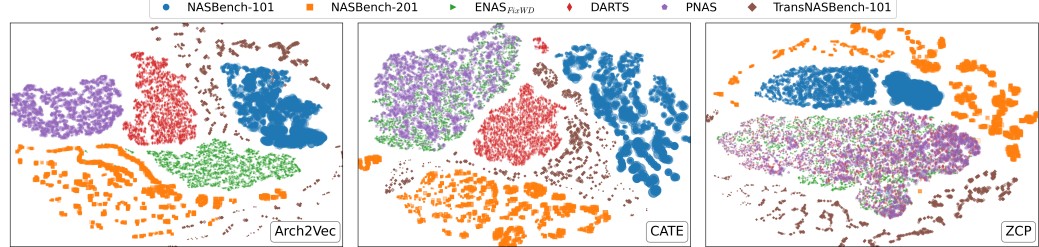

Figure 6: t-SNE scatterplot of the encodings for a set of architecture families using the ZCP, unified Arch2Vec and unified CATE encodings. Best viewed in color.

CIFAR-10, whereas NASBench-201 is trained on CIFAR-10, CIFAR-100, and ImageNet16-120. NASBench-301(Zela et al., 2020) serves as a surrogate NAS benchmark, containing a total of $10^{18}$ architectures. TransNAS-Bench-101(Duan et al., 2021) stands as a NAS benchmark that includes a micro (cell-based) search space with 4096 architectures and a macro search space embracing 3256 architectures. In our paper, we only study TransNASBench-101 Micro as that is a cell-based search space. These networks are individually trained on seven different tasks derived from the Taskonomy dataset. The NASLib framework unifies these search spaces. The NAS-Bench-Suite-Zero(Krishnakumar et al., 2022b) further extends this space by incorporating two datasets from NAS-Bench-360, SVHN, and another four datasets from Taskonomy. Further, the NDS(Radosavovic et al., 2019) spaces are described in Table 8 borrowed from the original paper. Additionally, the NDS data-set has '$FixWD$' data-sets which indicate that the width and depth do not vary in architectures. The NDS data-set has $LRWD$ data-sets which indicate that the learning rates do not vary in architectures. We do not include learning rate related representations in our predictor, while it is possible and may benefit performance. We only look at architectural aspects of the NAS design problem.

|  | num ops | num nodes | output | num cells (B) |
|---|---|---|---|---|
| NASNet (Zoph et al., 2018) | 13 | 5 | L | 71,465,842 |
| Amoeba (Real et al., 2019) | 8 | 5 | L | 556,628 |
| PNAS (Liu et al., 2018) | 8 | 5 | A | 556,628 |
| ENAS (Pham et al., 2018) | 5 | 5 | L | 5,063 |
| DARTS (Liu et al., 2019) | 8 | 4 | A | 242 |

Table 8: **NAS design spaces.** NDS (Radosavovic et al., 2019) summarizes the cell structure for five NAS design spaces. This table lists the number of candidate ops (5×5 conv, 3×3 max pool), number of nodes (excluding the inputs), and which nodes are concatenated for the output ('A' if 'all' nodes, 'L' if 'loose' nodes not used as input to other nodes). Given $o$ ops to choose from, there are $o^2 \cdot (j+1)^2$ choices when adding the $j^{th}$ node, leading to $o^{2k} \cdot ((k+1)!)^2$ possible cells with $k$ nodes (of course many of these cells are redundant). The spaces vary substantially; indeed, even exact candidate ops for each vary.

### A.3 ADDITIONAL RESULTS

In this sub-section, we provide more complete versions of some of the graphs in the main paper.

The t-SNE scatterplot showcased in Figure 6 demonstrates distinct clustering patterns associated with Arch2Vec based on different search spaces. This pattern is attributed to the binary indexing approach utilized in operations representation. Similar clustering tendencies are also observable for CATE and ZCP. However, it's noteworthy that search spaces like ENAS$FixWD$ and PNAS tend to cluster more closely in the CATE representation. This proximity is influenced by the similarities in their parameter counts. In the case of ZCP, DARTS shows a tendency to cluster within the ENAS$FixWD$ and PNAS spaces, which can be attributed to shared zero-cost characteristics. These observations highlight the distinct nature of the encoding methodologies employed by ZCP, Arch2Vec, and CATE. Quantitative analysis reveals the correlation of parameter count with the respective representations as 0.56 for ZCP, 0.38 for CATE, and 0.13 for Arch2Vec. This quantitative

| Samples | FLAN$^T$ | + Arch2Vec | + CATE | + ZCP | + CAZ | FLAN$^T$ | + Arch2Vec | + CATE | + ZCP | + CAZ |
|---|---|---|---|---|---|---|---|---|---|---|
| | **NASBench-201 $\longrightarrow$ NASBench-101** | | | | | **NASBench-101 $\longrightarrow$ NASBench-201** | | | | |
| 2 | 0.5252 | 0.2981 | **0.5320** | 0.4702 | 0.4438 | **0.6437** | 0.4835 | 0.4885 | 0.3996 | 0.5902 |
| 4 | 0.5119 | 0.4066 | **0.5178** | 0.4594 | 0.4236 | **0.6208** | 0.5236 | 0.4902 | 0.4369 | 0.5293 |
| 8 | **0.5361** | 0.4717 | 0.5127 | 0.5171 | 0.4861 | **0.6474** | 0.5675 | 0.6062 | 0.5197 | 0.6059 |
| 16 | **0.5803** | 0.5236 | 0.5628 | 0.5422 | 0.5344 | 0.6923 | 0.6451 | 0.6491 | 0.6409 | **0.7162** |
| | **ENAS $\longrightarrow$ DARTS** | | | | | **PNAS$_{FixWD}$ $\longrightarrow$ DARTS$_{FixWD}$** | | | | |
| 2 | 0.4946 | 0.5141 | 0.5050 | **0.5811** | 0.5464 | 0.4186 | 0.2941 | 0.4151 | **0.4548** | 0.3557 |
| 4 | 0.5039 | 0.4402 | 0.4789 | **0.5433** | 0.5135 | 0.3944 | 0.3120 | 0.3791 | **0.4400** | 0.4194 |
| 8 | 0.5061 | 0.5210 | 0.5254 | **0.5852** | 0.5336 | 0.3722 | 0.3534 | 0.3929 | **0.4320** | 0.3859 |
| 16 | 0.5150 | 0.4836 | 0.4845 | **0.5940** | 0.5526 | 0.3653 | 0.3698 | 0.3898 | **0.4363** | 0.4167 |
| | **DARTS $\longrightarrow$ ENAS** | | | | | **NASNet $\longrightarrow$ ENAS$_{FixWD}$** | | | | |
| 2 | 0.4526 | 0.4797 | 0.4442 | **0.5870** | 0.4391 | 0.3166 | 0.2474 | 0.4352 | **0.4566** | 0.4218 |
| 4 | 0.3980 | 0.3599 | 0.3916 | **0.4970** | 0.3743 | 0.2581 | 0.3001 | 0.3864 | **0.4471** | 0.3676 |
| 8 | 0.3908 | 0.4315 | 0.4094 | **0.5308** | 0.4184 | 0.2521 | 0.3984 | 0.3806 | **0.4602** | 0.4044 |
| 16 | 0.4323 | 0.4037 | 0.4337 | **0.5336** | 0.3688 | 0.3405 | 0.4126 | 0.3899 | **0.4361** | 0.4223 |
| | **Amoeba $\longrightarrow$ NASBench-101** | | | | | **Amoeba $\longrightarrow$ NASBench-201** | | | | |
| 2 | −0.1169 | 0.1210 | 0.1426 | **0.4690** | −0.0658 | 0.0727 | 0.2638 | 0.0058 | **0.4546** | 0.1907 |
| 4 | 0.0670 | 0.1529 | 0.1822 | **0.4185** | 0.1305 | 0.1156 | 0.2782 | 0.1853 | **0.3994** | 0.3091 |
| 8 | 0.2265 | 0.2734 | 0.2083 | **0.4113** | 0.2883 | 0.2704 | 0.4022 | 0.4071 | 0.3893 | **0.4343** |
| 16 | 0.2261 | 0.4310 | 0.2912 | **0.5072** | 0.3608 | 0.3858 | 0.4432 | 0.3893 | 0.4675 | **0.4691** |
| | **Amoeba $\longrightarrow$ DARTS** | | | | | **Amoeba $\longrightarrow$ DARTS$_{FixWD}$** | | | | |
| 2 | **0.5852** | 0.5269 | 0.5105 | 0.4769 | 0.5165 | 0.2606 | 0.1219 | 0.2261 | 0.3072 | **0.3215** |
| 4 | **0.5380** | 0.4891 | 0.4880 | 0.4788 | 0.5033 | 0.1904 | 0.1731 | 0.1606 | **0.2998** | 0.2833 |
| 8 | 0.5490 | 0.4660 | 0.5083 | 0.4825 | **0.5675** | 0.2481 | 0.2676 | 0.1469 | **0.3396** | 0.2983 |
| 16 | 0.5629 | 0.4929 | 0.5632 | 0.5531 | **0.5651** | 0.2230 | 0.2403 | 0.2354 | **0.3063** | 0.2816 |
| | **Amoeba $\longrightarrow$ ENAS** | | | | | **Amoeba $\longrightarrow$ ENAS$_{FixWD}$** | | | | |
| 2 | **0.5225** | 0.4255 | 0.5065 | 0.4745 | 0.3230 | 0.3063 | 0.2317 | 0.3965 | **0.4385** | 0.2940 |
| 4 | 0.4335 | 0.3989 | **0.4818** | 0.4763 | 0.3221 | 0.3127 | 0.2578 | 0.3942 | **0.4169** | 0.2258 |
| 8 | **0.4875** | 0.4200 | 0.4790 | 0.4841 | 0.3453 | 0.3602 | 0.3442 | 0.3899 | **0.3949** | 0.2809 |
| 16 | 0.4492 | 0.4529 | 0.4806 | **0.5027** | 0.3423 | 0.3508 | 0.2963 | 0.4166 | **0.4264** | 0.3310 |

Table 9: A study demonstrating the effectiveness of transferring predictors from the Amoeba NDS search-space to a target space using our Unified encodings. Table shows Kendall Tau correlation. 9 trials. Note that the first set of experiments have a different source space (as described by Source Space $\longrightarrow$ Target Space). This is to demonstrate that in cases where source and target spaces are similar, FLAN$^T$ may outperform other methods in the low sample count regime. In the same target spaces, when NDS Amoeba is used, supplementary encodings seem to aid transfer learning. This can be supported by observing that Amoeba has reduction and normal cells, with operation sets and macro-architecture quite distinct from NASBench-201 and NASBench-101. Additionally, we see that Amoeba to DARTS transfer does not necessarily benefit from supplementary encoding, whereas it does in DARTS$_{FixWD}$, this is because in both Amoeba and DARTS, the w-d vector changes, whereas DARTS$_{FixWD}$ is distinct in the sense that its w-d vector is fixed.

insight underscores the differential impact of encoding strategies on the parameter space representation across various search spaces.

## A.4 NEURAL ARCHITECTURE SEARCH ON NASBENCH-201 CIFAR-100

To demonstrate the effectiveness of our predictor on NAS on more search spaces, we compare FLAN with BRP-NAS to compare predictors, as well as other NAS search methodologies in Figure 7.

## A.5 ON RUN-TIME OF OUR PREDICTOR

Training FLAN is extremely efficient, with our median training time being approximately 7.5 minutes. This implies that modifications to search space descriptions or indexing can be done trivially and FLAN can be re-trained trivially. Further, generating the unified Arch2Vec and CATE encodings can both be done in under an hour on a consumer GPU. The time to transfer to a new search space depends upon the number of samples, our maximum time for transfer in tests was approximately

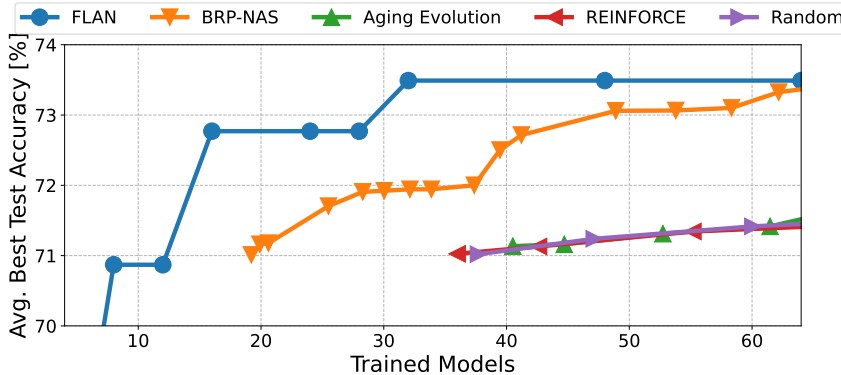

Figure 7: FLAN with the iterative sampling search algorithm (BRP-NAS) outperforms other popular search methodologies on NASBench-201 CIFAR100.

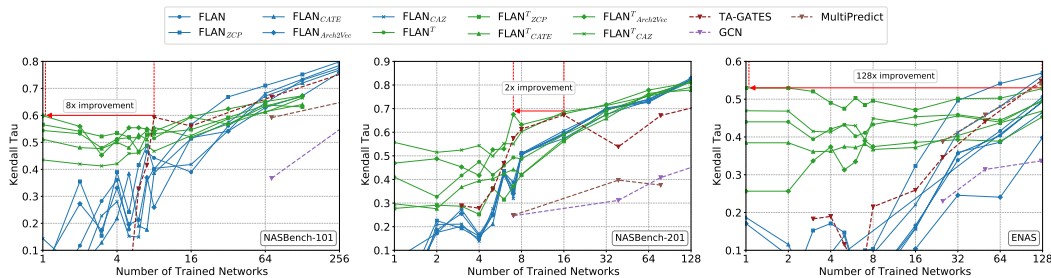

Figure 8: Prediction accuracy with different numbers of trained NNs. We investigate the impact of supplemental and unified encodings with FLAN, and compare to prior work. X-axis is logarithmic. Source space for NASBench-201 is NASBench-101 and vice versa. Source space for ENAS is DARTS.

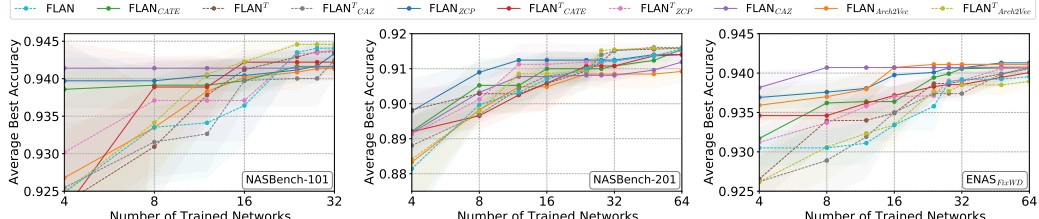

Figure 9: End-to-end NAS with different predictors using an iterative sampling search algorithm. $FLAN^T$ improves search efficiency in the low sample count region. Source search-space for NAS-Bench-201 is NASBench-101 and vice versa. Source space for $ENAS_{FixWD}$ is PNAS.
1 minute. Finally, for inference during NAS, we can evaluate approximately 160 architectures per second.

## A.6 NAS ON ALL SEARCH SPACES

In Figure 10 and Figure 11, we provide the NAS results a range of samples and representations on all 13 NAS spaces.

## A.7 EXPERIMENTAL SETUP

In this paper, we focus on standardizing our experiments on entire NAS Design spaces. We open source our code and generated encodings to foster further research. Additionally, we list the primary experimental hyperparameters in Table 12.

It is important to note that our results for the PATH encoding are generated with the naszilla hyperparamaters described in Table 13. Upon reproducing their set-up on our own MLP network archi-

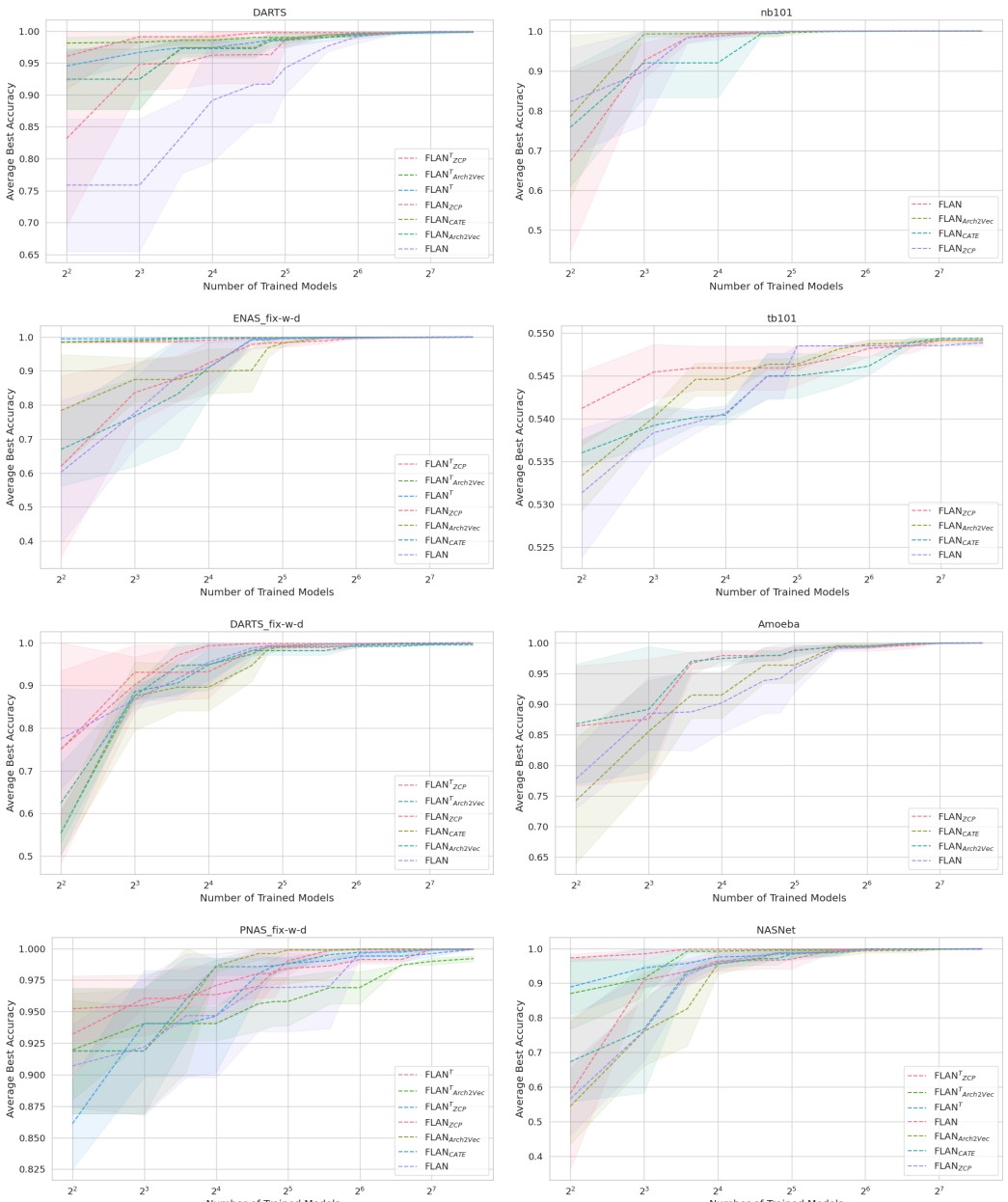

Figure 10: Neural Architecture Search on all NAS spaces detailed in the paper. Accuracies normalized 0-1.

tecture, adjacency representation was much better than path encoding. This further highlights the importance of predictor design. In Table 14, we see that their MetaNN ourperforms our NN design, but only for the PATH encoding.

## A.8    ARCHITECTURE DESIGN ABLATION

In this section, we take a deep look at key architectural decisions and how they impact the sample efficiency of predictors. Table 16 reproduces prior work (Ning et al., 2022) experimental setting and looks at the impact of 'Timesteps' (TS), 'Residual Connection' (RS), 'Zero Cost Symmetry Breaking' (ZCSB) and 'Architectural Zero Cost Proxy' (AZCP). We find that residual connection 'RS' has a major impact on KDT, causing a dip from 0.66 to 0.59 on the test indicated by 1% of NASBench-101. We extend this experimental setting to Table 9, where we study the impact of time-steps on **the entire search space**. We find that in 3 out of 4 searc hspaces, having

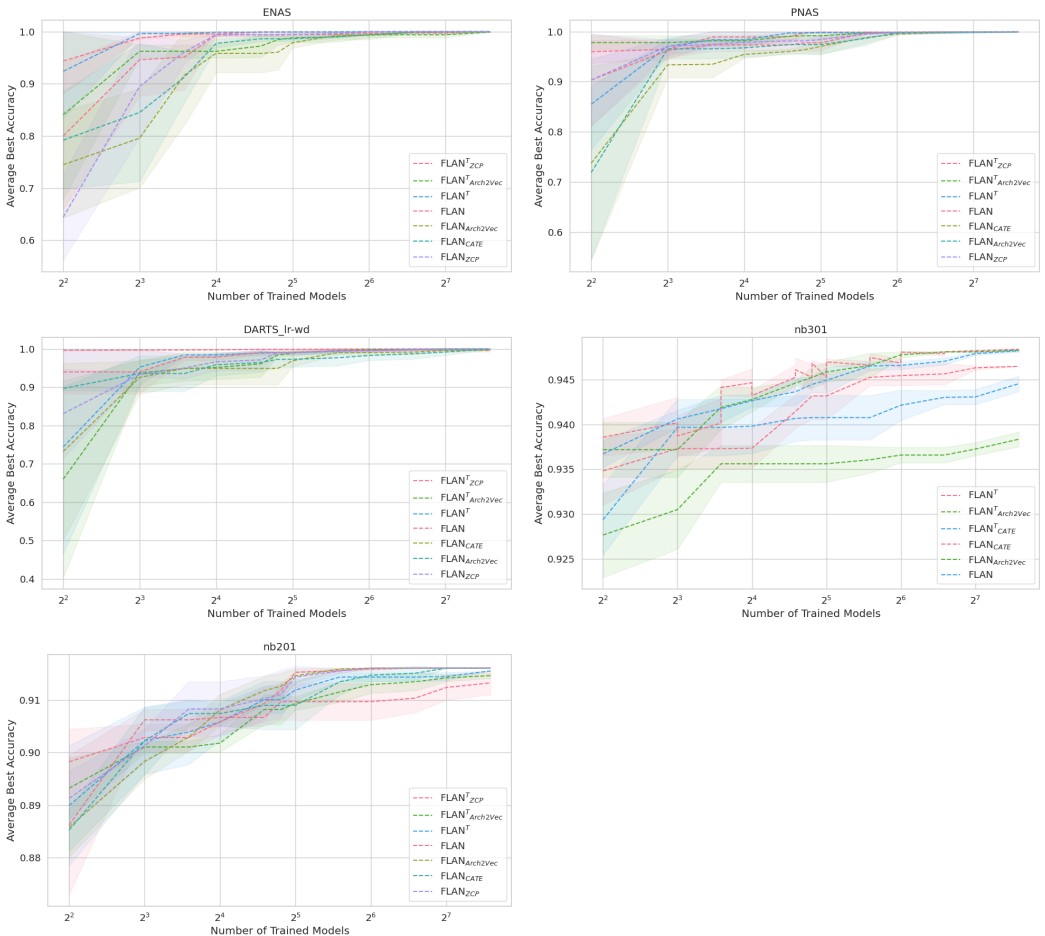

Figure 11: Neural Architecture Search on all NAS spaces detailed in the paper. Accuracies normalized 0-1 except NB201, NB301. Our NB301 does not have ZCP.

time-steps has a positive impact on accuracy. It is important to note that the impact is lesser than residual connection. This may indicate the importance of research in network design to avoid the over-smoothing problem in GCNs.

Finally, we conduct a large scale study on NASBench-101, NASBench-201 and PNAS. In this study, we look at the impact of having a 'DGF residual', 'GAT LeakyReLU' and 'GAT KQV-Projection'. From Equation 2, we can see that the projection matrix $W_p$ is shared, whereas in typical attention mechanism, we have different projection matrices for the key, query and value tensors.

Thus, using the 'KQV-Projection' implies using $W_{qp}$, $W_{kp}$ and $W_{vp}$ matrices as follows:

$$\text{Attn}_j(X^l) = \text{softmax}(\text{LeakyReLU}(A_j \cdot a(W_{qp}^l X^l \cdot W_{kp} X_j^l))) \cdot W_{vp} X_j^l \tag{4}$$

$$X^{l+1} = \text{LayerNorm}\left(\sigma(OW_o^l) \odot \sum_{j=1}^{n} \text{Attn}_j(X^l)\right) \tag{5}$$

### A.9 PREDICTOR SAMPLE EFFIENCY ON ALL SEARCH SPACES

In Table 10, we provide the kendall-tau for a range of samples and representations on all 13 NAS spaces.

| Space | Samples Representation | 2.0 | 4.0 | 5.0 | 7.0 | 8.0 | 16.0 | 32.0 | 64.0 | 128.0 |
|---|---|---|---|---|---|---|---|---|---|---|
| Amoeba | FLAN | $0.0084_{0.0003}$ | $0.0092_{0.0021}$ | $0.0883_{0.0008}$ | $0.0354_{0.0001}$ | $0.0468_{0.0003}$ | $0.0424_{0.0047}$ | $0.1578_{0.0116}$ | $0.2621_{0.0018}$ | $0.4237_{0.0003}$ |
| | $\text{FLAN}^T_{Arch2Vec}$ | $0.1253_{0.0005}$ | $0.0858_{0.0015}$ | $0.0841_{0.0019}$ | $0.0927_{0.0018}$ | $0.1255_{0.0008}$ | $0.1096_{0.0029}$ | $0.0972_{0.0052}$ | $0.1398_{0.0007}$ | $0.1973_{0.0018}$ |
| | $\text{FLAN}^T_{ZCP}$ | $\mathbf{0.2381_{0.0000}}$ | $\mathbf{0.1931_{0.0006}}$ | $\mathbf{0.2030_{0.0023}}$ | $\mathbf{0.2450_{0.0005}}$ | $\mathbf{0.2759_{0.0008}}$ | $\mathbf{0.2908_{0.0022}}$ | $0.3630_{0.0010}$ | $0.4140_{0.0012}$ | $0.4807_{0.0008}$ |
| | $\text{FLAN}_{Arch2Vec}$ | $0.0663_{0.0007}$ | $-0.0035_{0.0002}$ | $0.0705_{0.0003}$ | $0.0378_{0.0008}$ | $0.0526_{0.0001}$ | $0.0632_{0.0029}$ | $0.2147_{0.0181}$ | $0.2390_{0.0023}$ | $0.4386_{0.0009}$ |
| | $\text{FLAN}_{ZCP}$ | $-0.0182_{0.0033}$ | $0.0530_{0.0047}$ | $0.0927_{0.0005}$ | $0.0644_{0.0005}$ | $0.0944_{0.0005}$ | $0.1702_{0.0051}$ | $\mathbf{0.3689_{0.0131}}$ | $\mathbf{0.4259_{0.0006}}$ | $\mathbf{0.5552_{0.0001}}$ |
| DARTS | FLAN | $0.0061_{0.0236}$ | $0.0178_{0.0012}$ | $0.0119_{0.0000}$ | $0.0612_{0.0012}$ | $0.0208_{0.0039}$ | $0.0810_{0.0024}$ | $0.2380_{0.0045}$ | $0.4048_{0.0012}$ | $0.5108_{0.0003}$ |
| | $\text{FLAN}^T_{Arch2Vec}$ | $\mathbf{0.6267_{0.0000}}$ | $0.5765_{0.0000}$ | $\mathbf{0.5710_{0.0028}}$ | $0.5308_{0.0031}$ | $\mathbf{0.5661_{0.0008}}$ | $0.5792_{0.0004}$ | $0.5610_{0.0031}$ | $0.5927_{0.0018}$ | $0.5908_{0.0003}$ |
| | $\text{FLAN}^T_{ZCP}$ | $0.6033_{0.0000}$ | $\mathbf{0.6097_{0.0002}}$ | $0.5457_{0.0018}$ | $\mathbf{0.5905_{0.0011}}$ | $0.5413_{0.0078}$ | $\mathbf{0.6114_{0.0012}}$ | $\mathbf{0.5964_{0.0020}}$ | $\mathbf{0.6491_{0.0007}}$ | $\mathbf{0.6250_{0.0000}}$ |
| | $\text{FLAN}_{Arch2Vec}$ | $0.1581_{0.0000}$ | $0.0642_{0.0010}$ | $0.0148_{0.0005}$ | $0.0483_{0.0003}$ | $0.0138_{0.0033}$ | $0.0853_{0.0013}$ | $0.2513_{0.0030}$ | $0.3059_{0.0178}$ | $0.4780_{0.0006}$ |
| | $\text{FLAN}_{ZCP}$ | $0.0082_{0.0044}$ | $0.1904_{0.0196}$ | $0.0452_{0.0060}$ | $0.2233_{0.0096}$ | $0.0895_{0.0087}$ | $0.2392_{0.0016}$ | $0.4539_{0.0009}$ | $0.5458_{0.0014}$ | $0.5969_{0.0002}$ |
| $\text{DARTS}_{FixWD}$ | FLAN | $0.0553_{0.0007}$ | $0.0324_{0.0028}$ | $0.0547_{0.0030}$ | $0.0604_{0.0078}$ | $0.0994_{0.0031}$ | $0.0699_{0.0099}$ | $0.3758_{0.0001}$ | $0.4410_{0.0053}$ | $\mathbf{0.5642_{0.0001}}$ |
| | $\text{FLAN}^T_{Arch2Vec}$ | $0.3249_{0.0000}$ | $\mathbf{0.3064_{0.0015}}$ | $0.3362_{0.0011}$ | $0.2673_{0.0044}$ | $0.2820_{0.0087}$ | $0.2196_{0.0029}$ | $0.3803_{0.0025}$ | $0.3380_{0.0012}$ | $0.4048_{0.0016}$ |
| | $\text{FLAN}^T_{ZCP}$ | $\mathbf{0.3654_{0.0000}}$ | $0.2909_{0.0010}$ | $\mathbf{0.3473_{0.0002}}$ | $\mathbf{0.3589_{0.0013}}$ | $\mathbf{0.3975_{0.0002}}$ | $\mathbf{0.3756_{0.0000}}$ | $\mathbf{0.3959_{0.0005}}$ | $0.4001_{0.0009}$ | $0.4304_{0.0017}$ |
| | $\text{FLAN}_{Arch2Vec}$ | $0.0574_{0.0086}$ | $0.0582_{0.0011}$ | $0.0757_{0.0025}$ | $0.0477_{0.0007}$ | $0.1156_{0.0035}$ | $0.0756_{0.0024}$ | $0.3816_{0.0000}$ | $0.4432_{0.0057}$ | $0.5576_{0.0001}$ |
| | $\text{FLAN}_{ZCP}$ | $-0.0388_{0.0065}$ | $0.0845_{0.0006}$ | $0.0780_{0.0035}$ | $0.0772_{0.0004}$ | $0.1428_{0.0019}$ | $0.0930_{0.0088}$ | $0.3772_{0.0000}$ | $\mathbf{0.4969_{0.0025}}$ | $0.5542_{0.0016}$ |
| $\text{DARTS}_{LRWD}$ | FLAN | $0.0034_{0.0005}$ | $0.0123_{0.0001}$ | $0.0008_{0.0002}$ | $-0.0214_{0.0001}$ | $-0.0016_{0.0002}$ | $0.0056_{0.0004}$ | $0.0150_{0.0001}$ | $0.0336_{0.0005}$ | $0.0324_{0.0000}$ |
| | $\text{FLAN}^T_{Arch2Vec}$ | $\mathbf{0.1335_{0.0000}}$ | $0.0912_{0.0011}$ | $0.0802_{0.0005}$ | $0.1169_{0.0000}$ | $\mathbf{0.1270_{0.0017}}$ | $\mathbf{0.0898_{0.0003}}$ | $0.0584_{0.0003}$ | $0.0534_{0.0053}$ | $0.0320_{0.0077}$ |
| | $\text{FLAN}^T_{ZCP}$ | $0.1212_{0.0000}$ | $\mathbf{0.1020_{0.0035}}$ | $\mathbf{0.0972_{0.0016}}$ | $0.0372_{0.0013}$ | $0.1001_{0.0004}$ | $0.0865_{0.0005}$ | $\mathbf{0.0927_{0.0020}}$ | $0.0877_{0.0009}$ | $0.0244_{0.0026}$ |
| | $\text{FLAN}_{Arch2Vec}$ | $0.0300_{0.0000}$ | $0.0058_{0.0001}$ | $0.0033_{0.0001}$ | $-0.0226_{0.0001}$ | $-0.0019_{0.0001}$ | $0.0026_{0.0005}$ | $0.0102_{0.0002}$ | $0.0366_{0.0003}$ | $0.0373_{0.0001}$ |
| | $\text{FLAN}_{ZCP}$ | $0.0035_{0.0003}$ | $0.0072_{0.0000}$ | $0.0058_{0.0000}$ | $-0.0033_{0.0006}$ | $-0.0021_{0.0001}$ | $0.0294_{0.0011}$ | $0.0292_{0.0007}$ | $0.0474_{0.0004}$ | $0.0464_{0.0002}$ |
| ENAS | FLAN | $-0.0276_{0.0020}$ | $0.0729_{0.0002}$ | $0.0114_{0.0006}$ | $0.0216_{0.0006}$ | $0.0147_{0.0014}$ | $0.1627_{0.0108}$ | $0.3430_{0.0018}$ | $0.4043_{0.0008}$ | $0.4753_{0.0008}$ |
| | $\text{FLAN}^T_{Arch2Vec}$ | $0.4568_{0.0000}$ | $\mathbf{0.4598_{0.0008}}$ | $0.4521_{0.0016}$ | $0.4614_{0.0017}$ | $0.4278_{0.0002}$ | $0.4130_{0.0043}$ | $0.4565_{0.0032}$ | $0.4820_{0.0013}$ | $0.5146_{0.0001}$ |
| | $\text{FLAN}^T_{ZCP}$ | $\mathbf{0.5255_{0.0000}}$ | $0.3646_{0.0149}$ | $\mathbf{0.5331_{0.0002}}$ | $\mathbf{0.4882_{0.0013}}$ | $\mathbf{0.4695_{0.0055}}$ | $\mathbf{0.5129_{0.0003}}$ | $\mathbf{0.5254_{0.0005}}$ | $0.5296_{0.0009}$ | $0.5411_{0.0003}$ |
| | $\text{FLAN}_{Arch2Vec}$ | $0.1154_{0.0121}$ | $0.0320_{0.0006}$ | $0.0428_{0.0016}$ | $0.0108_{0.0003}$ | $0.0016_{0.0042}$ | $0.1583_{0.0183}$ | $0.3620_{0.0037}$ | $0.3851_{0.0028}$ | $0.4594_{0.0012}$ |
| | $\text{FLAN}_{ZCP}$ | $-0.0192_{0.0031}$ | $0.1497_{0.0002}$ | $0.1323_{0.0055}$ | $0.1188_{0.0077}$ | $0.1206_{0.0054}$ | $0.3272_{0.0239}$ | $0.4951_{0.0002}$ | $\mathbf{0.5416_{0.0024}}$ | $\mathbf{0.5714_{0.0003}}$ |
| $\text{ENAS}_{FixWD}$ | FLAN | $0.0319_{0.0026}$ | $-0.0274_{0.0010}$ | $0.0570_{0.0120}$ | $0.1213_{0.0191}$ | $0.0604_{0.0046}$ | $0.1298_{0.0092}$ | $0.3625_{0.0118}$ | $0.5038_{0.0002}$ | $0.5425_{0.0002}$ |
| | $\text{FLAN}^T_{Arch2Vec}$ | $0.4006_{0.0000}$ | $0.3597_{0.0025}$ | $0.3063_{0.0234}$ | $0.3861_{0.0021}$ | $0.2657_{0.0119}$ | $\mathbf{0.4055_{0.0006}}$ | $0.3480_{0.0070}$ | $0.3314_{0.0006}$ | $0.4176_{0.0000}$ |
| | $\text{FLAN}^T_{ZCP}$ | $\mathbf{0.4282_{0.0000}}$ | $\mathbf{0.3996_{0.0001}}$ | $\mathbf{0.4621_{0.0011}}$ | $\mathbf{0.4130_{0.0003}}$ | $\mathbf{0.4389_{0.0013}}$ | $0.3804_{0.0050}$ | $0.3947_{0.0000}$ | $0.4315_{0.0002}$ | $0.4915_{0.0002}$ |
| | $\text{FLAN}_{Arch2Vec}$ | $0.0375_{0.0004}$ | $-0.0345_{0.0032}$ | $0.0627_{0.0096}$ | $0.1116_{0.0149}$ | $0.0072_{0.0050}$ | $0.1417_{0.0093}$ | $0.3401_{0.0148}$ | $0.5091_{0.0001}$ | $0.5464_{0.0006}$ |
| | $\text{FLAN}_{ZCP}$ | $-0.0662_{0.0029}$ | $-0.0340_{0.0062}$ | $0.0732_{0.0013}$ | $0.2049_{0.0050}$ | $0.1145_{0.0084}$ | $0.2216_{0.0087}$ | $\mathbf{0.4333_{0.0037}}$ | $\mathbf{0.5231_{0.0003}}$ | $\mathbf{0.5511_{0.0001}}$ |
| NASNet | FLAN | $-0.0322_{0.0002}$ | $0.0669_{0.0016}$ | $0.0415_{0.0014}$ | $0.0561_{0.0027}$ | $0.0373_{0.0001}$ | $0.0543_{0.0006}$ | $0.1997_{0.0005}$ | $0.2726_{0.0028}$ | $0.4090_{0.0001}$ |
| | $\text{FLAN}^T_{Arch2Vec}$ | $0.3539_{0.0000}$ | $0.3421_{0.0004}$ | $0.3319_{0.0027}$ | $0.3846_{0.0005}$ | $0.2875_{0.0018}$ | $0.2989_{0.0038}$ | $0.3109_{0.0006}$ | $0.3466_{0.0005}$ | $0.3981_{0.0002}$ |
| | $\text{FLAN}^T_{ZCP}$ | $\mathbf{0.3825_{0.0000}}$ | $\mathbf{0.3732_{0.0049}}$ | $\mathbf{0.4137_{0.0041}}$ | $\mathbf{0.4033_{0.0003}}$ | $\mathbf{0.3893_{0.0019}}$ | $\mathbf{0.4115_{0.0014}}$ | $\mathbf{0.4607_{0.0020}}$ | $0.4268_{0.0007}$ | $0.5020_{0.0001}$ |
| | $\text{FLAN}_{Arch2Vec}$ | $0.0512_{0.0007}$ | $0.0693_{0.0011}$ | $0.0426_{0.0002}$ | $0.0564_{0.0126}$ | $0.0432_{0.0010}$ | $0.0444_{0.0015}$ | $0.2186_{0.0011}$ | $0.2664_{0.0006}$ | $0.4411_{0.0001}$ |
| | $\text{FLAN}_{ZCP}$ | $-0.0510_{0.0040}$ | $0.1379_{0.0029}$ | $0.0821_{0.0001}$ | $0.1532_{0.0203}$ | $0.0671_{0.0102}$ | $0.2126_{0.0059}$ | $0.3770_{0.0069}$ | $\mathbf{0.4602_{0.0014}}$ | $\mathbf{0.5364_{0.0015}}$ |
| PNAS | FLAN | $0.0215_{0.0086}$ | $0.0046_{0.0030}$ | $0.0542_{0.0004}$ | $-0.0085_{0.0042}$ | $0.0248_{0.0002}$ | $0.0268_{0.0006}$ | $0.1617_{0.0022}$ | $0.3260_{0.0064}$ | $0.4997_{0.0004}$ |
| | $\text{FLAN}^T_{Arch2Vec}$ | $0.0790_{0.0000}$ | $0.0954_{0.0239}$ | $0.0849_{0.0064}$ | $0.1317_{0.0241}$ | $0.2432_{0.0023}$ | $0.1137_{0.0036}$ | $0.2365_{0.0056}$ | $0.3105_{0.0012}$ | $0.3482_{0.0022}$ |
| | $\text{FLAN}^T_{ZCP}$ | $0.0408_{0.0000}$ | $0.1555_{0.0085}$ | $0.1087_{0.0017}$ | $0.1511_{0.0069}$ | $0.2167_{0.0014}$ | $0.1789_{0.0021}$ | $0.2847_{0.0022}$ | $0.3430_{0.0003}$ | $0.3725_{0.0034}$ |
| | $\text{FLAN}_{Arch2Vec}$ | $0.1010_{0.0000}$ | $0.0116_{0.0036}$ | $0.0538_{0.0009}$ | $-0.0157_{0.0035}$ | $0.0354_{0.0008}$ | $0.0294_{0.0006}$ | $0.1801_{0.0036}$ | $0.2702_{0.0042}$ | $0.4506_{0.0007}$ |
| | $\text{FLAN}_{ZCP}$ | $-0.0159_{0.0024}$ | $0.1115_{0.0081}$ | $0.0456_{0.0001}$ | $0.0373_{0.0051}$ | $0.0456_{0.0003}$ | $0.1675_{0.0052}$ | $\mathbf{0.3856_{0.0024}}$ | $\mathbf{0.4564_{0.0011}}$ | $\mathbf{0.5298_{0.0001}}$ |
| $\text{PNAS}_{FixWD}$ | FLAN | $0.0806_{0.0024}$ | $0.0550_{0.0011}$ | $0.0654_{0.0032}$ | $0.1131_{0.0011}$ | $0.1287_{0.0065}$ | $\mathbf{0.3268_{0.0008}}$ | $0.3815_{0.0075}$ | $0.4669_{0.0005}$ | $0.5896_{0.0008}$ |
| | $\text{FLAN}^T_{Arch2Vec}$ | $0.1102_{0.0000}$ | $0.1410_{0.0016}$ | $0.1427_{0.0009}$ | $0.0927_{0.0002}$ | $0.1225_{0.0102}$ | $0.1340_{0.0027}$ | $0.2003_{0.0003}$ | $0.2068_{0.0048}$ | $0.3231_{0.0026}$ |
| | $\text{FLAN}^T_{ZCP}$ | $\mathbf{0.3273_{0.0000}}$ | $0.1594_{0.0196}$ | $\mathbf{0.3085_{0.0008}}$ | $\mathbf{0.3203_{0.0026}}$ | $\mathbf{0.2778_{0.0007}}$ | $0.2169_{0.0004}$ | $0.3030_{0.0012}$ | $0.3259_{0.0010}$ | $0.3422_{0.0026}$ |
| | $\text{FLAN}_{Arch2Vec}$ | $0.0063_{0.0054}$ | $0.0739_{0.0020}$ | $0.0422_{0.0005}$ | $0.1308_{0.0025}$ | $0.0703_{0.0061}$ | $0.2923_{0.0041}$ | $0.2928_{0.0016}$ | $0.4585_{0.0006}$ | $0.5816_{0.0007}$ |
| | $\text{FLAN}_{ZCP}$ | $-0.0304_{0.0058}$ | $0.1099_{0.0013}$ | $0.0376_{0.0036}$ | $0.1227_{0.0007}$ | $0.0805_{0.0062}$ | $0.2928_{0.0016}$ | $\mathbf{0.3565_{0.0022}}$ | $\mathbf{0.4752_{0.0002}}$ | $\mathbf{0.5950_{0.0001}}$ |
| NB101 | FLAN | $0.1206_{0.0452}$ | $0.3525_{0.0150}$ | $0.4118_{0.0007}$ | $0.4327_{0.0036}$ | $0.4572_{0.0014}$ | $0.4657_{0.0067}$ | $0.5692_{0.0017}$ | $\mathbf{0.6905_{0.0000}}$ | $0.7339_{0.0002}$ |
| | $\text{FLAN}^T_{Arch2Vec}$ | $0.3566_{0.0002}$ | $0.3542_{0.0001}$ | $0.3757_{0.0016}$ | $0.3764_{0.0027}$ | $0.3443_{0.0018}$ | $0.3990_{0.0003}$ | $0.4768_{0.0015}$ | $0.5561_{0.0004}$ | $0.5917_{0.0011}$ |
| | $\text{FLAN}^T_{ZCP}$ | $\mathbf{0.5432_{0.0001}}$ | $0.4949_{0.0033}$ | $\mathbf{0.5517_{0.0023}}$ | $\mathbf{0.5852_{0.0003}}$ | $\mathbf{0.5536_{0.0003}}$ | $\mathbf{0.5989_{0.0002}}$ | $\mathbf{0.6438_{0.0006}}$ | $0.6651_{0.0005}$ | $0.6932_{0.0003}$ |
| | $\text{FLAN}_{Arch2Vec}$ | $-0.0748_{0.0899}$ | $0.0009_{0.0781}$ | $0.4006_{0.0035}$ | $0.1432_{0.0488}$ | $0.3256_{0.0531}$ | $0.4549_{0.0017}$ | $0.5713_{0.0013}$ | $0.6679_{0.0002}$ | $0.7248_{0.0002}$ |
| | $\text{FLAN}_{ZCP}$ | $-0.0311_{0.0312}$ | $\mathbf{0.4972_{0.0004}}$ | $0.2995_{0.0059}$ | $0.4604_{0.0026}$ | $0.3095_{0.0566}$ | $0.5452_{0.0024}$ | $0.5995_{0.0004}$ | $0.6789_{0.0012}$ | $\mathbf{0.7567_{0.0001}}$ |
| NB201 | FLAN | $0.1714_{0.0227}$ | $0.1610_{0.0830}$ | $0.2600_{0.0828}$ | $0.3776_{0.0076}$ | $0.5102_{0.0005}$ | $0.5928_{0.0018}$ | $0.6991_{0.0008}$ | $0.7346_{0.0005}$ | $\mathbf{0.8291_{0.0002}}$ |
| | $\text{FLAN}^T_{CATE}$ | $0.1214_{0.0573}$ | $\mathbf{0.5467_{0.0245}}$ | $\mathbf{0.5306_{0.0035}}$ | $\mathbf{0.5284_{0.0013}}$ | $\mathbf{0.5179_{0.0016}}$ | $\mathbf{0.6352_{0.0033}}$ | $0.6226_{0.0033}$ | $0.7385_{0.0004}$ | $0.7885_{0.0006}$ |
| | $\text{FLAN}^T_{Arch2Vec}$ | $0.1444_{0.0049}$ | $-0.0201_{0.0058}$ | $0.2747_{0.0184}$ | $0.2291_{0.0029}$ | $0.3759_{0.0254}$ | $0.5254_{0.0102}$ | $0.6207_{0.0031}$ | $0.6879_{0.0003}$ | $0.7768_{0.0007}$ |
| | $\text{FLAN}_{Arch2Vec}$ | $0.1752_{0.0101}$ | $0.1488_{0.0951}$ | $0.2165_{0.1043}$ | $0.3648_{0.0077}$ | $0.5079_{0.0014}$ | $0.5945_{0.0018}$ | $\mathbf{0.7025_{0.0094}}$ | $0.7343_{0.0005}$ | $0.8206_{0.0001}$ |
| | $\text{FLAN}_{ZCP}$ | $\mathbf{0.2255_{0.0284}}$ | $0.1455_{0.1144}$ | $0.2505_{0.0891}$ | $0.3441_{0.0128}$ | $0.5152_{0.0005}$ | $0.5646_{0.0072}$ | $0.6723_{0.0026}$ | $0.7318_{0.0008}$ | $0.8259_{0.0001}$ |
| NB301 | FLAN | $\mathbf{0.4288_{0.0029}}$ | $\mathbf{0.2676_{0.0521}}$ | $\mathbf{0.4834_{0.0000}}$ | $\mathbf{0.4783_{0.0081}}$ | $\mathbf{0.5309_{0.0041}}$ | $\mathbf{0.5506_{0.0009}}$ | $\mathbf{0.6582_{0.0001}}$ | $\mathbf{0.7257_{0.0008}}$ | $\mathbf{0.7390_{0.0005}}$ |
| | $\text{FLAN}^T_{CATE}$ | $0.0241_{0.0135}$ | $0.0439_{0.0276}$ | $0.0706_{0.0319}$ | $0.1533_{0.0174}$ | $0.1221_{0.0291}$ | $0.3082_{0.0050}$ | $0.5230_{0.0021}$ | $0.6090_{0.0020}$ | $0.7328_{0.0003}$ |
| | $\text{FLAN}^T_{Arch2Vec}$ | $0.2735_{0.0002}$ | $0.1940_{0.0031}$ | $0.1574_{0.0068}$ | $0.2641_{0.0372}$ | $0.3843_{0.0022}$ | $0.4318_{0.0047}$ | $0.4993_{0.0028}$ | $0.6906_{0.0015}$ | $0.7205_{0.0005}$ |
| TB101 | FLAN | $0.0016_{0.0084}$ | $0.4824_{0.0042}$ | $0.4214_{0.0513}$ | $\mathbf{0.6398_{0.0001}}$ | $0.5475_{0.0109}$ | $\mathbf{0.6689_{0.0012}}$ | | $0.7150_{0.0002}$ | $0.7520_{0.0007}$ |
| | $\text{FLAN}^T_{ZCP}$ | $0.1862_{0.0099}$ | $0.4002_{0.0045}$ | $0.5109_{0.0086}$ | $0.5710_{0.0022}$ | $\mathbf{0.6201_{0.0000}}$ | $0.5599_{0.0037}$ | $\mathbf{0.6544_{0.0002}}$ | $0.7086_{0.0023}$ | $0.7298_{0.0015}$ |
| | $\text{FLAN}_{Arch2Vec}$ | $\mathbf{0.4061_{0.0075}}$ | $0.3605_{0.0778}$ | $\mathbf{0.5284_{0.0021}}$ | $0.5506_{0.0002}$ | $0.4597_{0.0218}$ | $0.5556_{0.0071}$ | $0.6364_{0.0038}$ | $0.7143_{0.0015}$ | $0.7670_{0.0000}$ |
| | $\text{FLAN}_{ZCP}$ | $0.3960_{0.0183}$ | $0.3793_{0.0750}$ | $0.5204_{0.0022}$ | $0.5591_{0.0010}$ | $0.4145_{0.0137}$ | $0.5381_{0.0052}$ | $0.6489_{0.0036}$ | $\mathbf{0.7219_{0.0011}}$ | $\mathbf{0.7747_{0.0001}}$ |

Table 10: Predictor Sample Efficiency of all NAS spaces detailed in the paper. 3 trials.

| Source | NB201 | NB301 | NB101 | NB201 | $\text{PNAS}_{FixWD}$ | $\text{ENAS}_{FixWD}$ | |
|---|---|---|---|---|---|---|---|
| Target | NB101 | NB301 | NB301 | TB101 | Amoeba | $\text{PNAS}_{FixWD}$ | |
| Source | NASNet | DARTS | ENAS | PNAS | $\text{DARTS}_{LRWD}$ | $\text{DARTS}_{FixWD}$ | PNAS |
| Target | $\text{ENAS}_{FixWD}$ | NASNet | DARTS | ENAS | PNAS | $\text{DARTS}_{LRWD}$ | $\text{DARTS}_{FixWD}$ |

Table 11: Source and Target Spaces for Table 10

| Hyperparameter | Value | Hyperparameter | Value |
|---|---|---|---|
| Learning Rate | 0.001 | Weight Decay | 0.00001 |
| Number of Epochs | 150 | Batch Size | 8 |
| Number of Transfer Epochs | 30 | Transfer Learning Rate | 0.001 |
| Graph Type | 'DGF+GAT ensemble' | Op Embedding Dim | 48 |
| Node Embedding Dim | 48 | Hidden Dim | 96 |
| GCN Dims | [128, 128, 128, 128, 128] | MLP Dims | [200, 200, 200] |
| GCN Output Conversion MLP | [128, 128] | Backward GCN Out Dims | [128, 128, 128, 128, 128] |
| OpEmb Update MLP Dims | [128] | NN Emb Dims | 128 |
| Supplementary Encoding Embedder Dims | [128, 128] | Number of Time Steps | 2 |
| Number of Trials | 9 | Loss Type | Pairwise Hinge Loss (Ning et al., 2022) |

Table 12: Hyperparameters used in main table experiments.

## A.10 RESULTS WITH VARIANCE

In this subsection, we attach results with variance for each experiment. Table **??** compares our method with existing encoders. Table 19 extends this to the entire search space and highlights the

| Parameter | Value | Parameter | Value |
|---|---|---|---|
| Loss | MAE | NN Depth | 10 |
| NN Width | 20 | Epochs | 200 |
| Batch Size | 32 | LR | 0.01 |

Table 13: Hyperparameters used to generate PATH results.

| Training Samples | Adj MetaNN | Adj NN | Path MetaNN | Path NN |
|---|---|---|---|---|
| 72 | 0.057 | **0.3270** | **0.3875** | -0.0315 |
| 364 | 0.1464 | **0.4647** | **0.6967** | -0.0363 |
| 729 | 0.2269 | **0.5141** | **0.7524** | -0.0023 |

Table 14: Study on PATH Encoding for NASBench-101. Tested on 7290 samples.

| | | | | NASBench-101 | | |
|---|---|---|---|---|---|---|
| Timesteps | DGF Residual | Leaky ReLU | KQV Projection | \multicolumn{3}{c}{Number Of Samples} |
| | | | | 8 | 16 | 32 |
| 1 | ✓ | ✗ | ✓ | 0.3945 | 0.4939 | 0.5340 |
| 1 | ✗ | ✓ | ✓ | 0.2425 | 0.4434 | 0.5448 |
| 1 | ✓ | ✓ | ✓ | 0.3348 | 0.5230 | 0.5829 |
| 1 | ✓ | ✓ | ✗ | 0.4129 | 0.5301 | 0.5442 |
| 1 | ✗ | ✗ | ✓ | 0.3132 | 0.4311 | 0.5454 |
| 1 | ✗ | ✗ | ✗ | **0.4791** | 0.4658 | 0.5123 |
| 1 | ✗ | ✓ | ✗ | 0.4595 | 0.5098 | **0.5904** |
| 2 | ✓ | ✗ | ✗ | 0.4628 | 0.5299 | 0.4825 |
| 2 | ✗ | ✓ | ✓ | 0.4487 | 0.4832 | 0.5582 |
| 2 | ✗ | ✓ | ✗ | 0.3403 | **0.5428** | 0.5495 |
| 2 | ✗ | ✗ | ✓ | 0.3562 | 0.4420 | 0.4737 |
| 2 | ✗ | ✗ | ✗ | 0.2640 | 0.5162 | 0.5316 |
| 2 | ✓ | ✓ | ✓ | 0.3939 | 0.5081 | 0.5684 |
| 3 | ✓ | ✓ | ✗ | 0.3899 | 0.4633 | 0.5446 |
| 3 | ✓ | ✓ | ✓ | 0.4756 | 0.5340 | 0.5484 |
| 3 | ✓ | ✗ | ✗ | 0.3020 | 0.5025 | 0.5616 |
| 3 | ✗ | ✓ | ✗ | 0.2957 | 0.3765 | 0.5607 |
| 3 | ✗ | ✓ | ✓ | 0.3280 | 0.4647 | 0.5552 |
| 3 | ✗ | ✗ | ✓ | 0.3674 | 0.5241 | 0.5291 |

Table 15: Results of architecture design ablation. Tested on 1000 randomly sampled architectures. Average over 3 trials. Results depict the Kendall Tau Correlation of FLAN with different DGF GAT module implementations.

| | \multicolumn{8}{c}{1% of NASBench-101} | \multicolumn{7}{c}{5% of NASBench-101} |
|---|---|---|---|---|---|---|---|---|---|---|---|---|---|---|---|
| TS | 1 | 2 | 3 | 1 | 2 | 2 | 2 | 2 | 1 | 2 | 3 | 1 | 1 | 1 | 1 |
| RS | ✓ | ✓ | ✓ | ✓ | ✓ | ✓ | ✓ | ✗ | ✓ | ✓ | ✓ | ✓ | ✓ | ✓ | ✓ |
| ZCSB | ✓ | ✓ | ✓ | ✓ | ✓ | ✗ | ✗ | ✓ | ✓ | ✓ | ✓ | ✗ | ✗ | ✗ | ✗ |
| AZCP | ✗ | ✗ | ✗ | ✓ | ✓ | ✓ | ✗ | ✓ | ✗ | ✗ | ✗ | ✓ | ✗ | ✓ | ✓ |
| KDT | 0.65 | 0.67 | 0.66 | **0.68** | 0.66 | 0.65 | 0.65 | 0.59 | 0.78 | 0.76 | 0.76 | **0.79** | 0.76 | 0.78 | 0.78 |

Table 16: Ablation for training on x% of 7290 samples on NB101, and testing on 7290 samples on NB101.

| Timesteps | $DARTS_{FixWD}$ | $ENAS_{FixWD}$ | NB101 | TB101 |
|---|---|---|---|---|
| 1 | $\mathbf{0.4870_{0.0002}}$ | $0.4653_{0.0031}$ | $0.7017_{0.0007}$ | $0.7789_{0.0002}$ |
| 2 | $0.4632_{0.0003}$ | $0.4799_{0.0010}$ | $0.7129_{0.0001}$ | $\mathbf{0.7939_{0.0001}}$ |
| 4 | $0.4801_{0.0001}$ | $\mathbf{0.4803_{0.0012}}$ | $\mathbf{0.7133_{0.0001}}$ | $0.7907_{0.0002}$ |

Table 17: We study the importance of time-steps in the FLAN predictor design. 128 samples are used to train, and tested on the entire NAS space.

effectiveness of adding supplementary encodings. Table 20 highlights the transfer results across search spaces, and Table 21 looks at module design for FLAN.

| | | NASBench-201 | | Number Of Samples | | | | | PNAS | | Number Of Samples | | |
|---|---|---|---|---|---|---|---|---|---|---|---|---|---|
| Timesteps | DGF Residual | Leaky ReLU | KQV Projection | 8 | 16 | 32 | Timesteps | DGF Residual | Leaky ReLU | KQV Projection | 8 | 16 | 32 |
| 1 | ✓ | ✗ | ✓ | 0.5550 | 0.6265 | 0.6850 | 1 | ✓ | ✗ | ✗ | 0.1887 | 0.3354 | 0.4763 |
| 1 | ✓ | ✗ | ✗ | 0.5415 | 0.6074 | 0.6767 | 1 | ✓ | ✗ | ✓ | 0.1436 | 0.2878 | 0.3994 |
| 1 | ✗ | ✓ | ✓ | 0.5425 | 0.6127 | 0.6841 | 1 | ✗ | ✗ | ✓ | 0.1612 | 0.3129 | 0.4290 |
| 1 | ✓ | ✓ | ✓ | 0.5437 | 0.6115 | 0.6830 | 1 | ✗ | ✓ | ✓ | 0.1526 | 0.2919 | 0.4178 |
| 1 | ✗ | ✗ | ✓ | 0.5529 | 0.6200 | 0.6773 | 1 | ✗ | ✗ | ✗ | 0.2085 | 0.3242 | 0.4546 |
| 1 | ✓ | ✓ | ✗ | 0.5563 | 0.6100 | 0.6766 | 1 | ✓ | ✓ | ✗ | 0.1823 | 0.3417 | **0.4837** |
| 1 | ✗ | ✗ | ✗ | 0.5460 | 0.5883 | 0.6886 | 1 | ✓ | ✓ | ✓ | 0.1472 | 0.3096 | 0.4467 |
| 1 | ✗ | ✓ | ✗ | 0.5303 | 0.5864 | 0.6886 | 1 | ✗ | ✗ | ✗ | 0.2031 | 0.3121 | 0.4540 |
| 2 | ✓ | ✓ | ✓ | 0.5295 | 0.6173 | 0.6796 | 2 | ✓ | ✓ | ✗ | 0.2417 | 0.3568 | 0.4666 |
| 2 | ✓ | ✓ | ✗ | 0.5431 | 0.6025 | 0.6847 | 2 | ✓ | ✗ | ✗ | 0.2448 | 0.3617 | 0.4662 |
| 2 | ✓ | ✗ | ✓ | 0.5452 | 0.6284 | 0.6800 | 2 | ✓ | ✗ | ✓ | 0.1804 | 0.3212 | 0.4760 |
| 2 | ✓ | ✗ | ✗ | 0.5545 | 0.5993 | 0.6758 | 2 | ✗ | ✓ | ✗ | 0.2337 | 0.3448 | 0.4397 |
| 2 | ✗ | ✓ | ✓ | 0.5512 | 0.6204 | 0.6781 | 2 | ✓ | ✗ | ✗ | 0.1822 | 0.3238 | 0.4583 |
| 2 | ✗ | ✓ | ✗ | 0.5396 | 0.5906 | 0.6807 | 2 | ✗ | ✗ | ✗ | 0.2183 | 0.3263 | 0.4368 |
| 2 | ✗ | ✗ | ✓ | 0.5280 | 0.6207 | 0.6781 | 2 | ✗ | ✗ | ✓ | 0.1912 | 0.3091 | 0.4516 |
| 2 | ✗ | ✗ | ✗ | 0.5470 | 0.5945 | **0.6956** | 2 | ✗ | ✗ | ✓ | 0.1866 | 0.3125 | 0.4406 |
| 3 | ✓ | ✓ | ✓ | 0.5488 | 0.6314 | 0.6849 | 3 | ✓ | ✗ | ✗ | 0.2523 | **0.3639** | 0.4541 |
| 3 | ✓ | ✓ | ✗ | **0.5644** | 0.6058 | 0.6751 | 3 | ✗ | ✗ | ✗ | 0.2271 | 0.3511 | 0.4496 |
| 3 | ✓ | ✗ | ✗ | 0.5529 | 0.5994 | 0.6806 | 3 | ✗ | ✗ | ✓ | 0.1802 | 0.3199 | 0.4477 |
| 3 | ✓ | ✗ | ✓ | 0.5457 | **0.6340** | 0.6863 | 3 | ✓ | ✓ | ✗ | **0.2586** | 0.3488 | 0.4606 |
| 3 | ✗ | ✓ | ✗ | 0.5579 | 0.5999 | 0.6909 | 3 | ✗ | ✓ | ✗ | 0.2100 | 0.3438 | 0.4442 |
| 3 | ✗ | ✗ | ✓ | 0.5429 | 0.6321 | 0.6874 | 3 | ✗ | ✓ | ✓ | 0.1814 | 0.3107 | 0.4415 |
| 3 | ✗ | ✓ | ✓ | 0.5372 | 0.6279 | 0.6835 | 3 | ✗ | ✗ | ✓ | 0.1770 | 0.3316 | 0.4660 |
| 3 | ✗ | ✗ | ✗ | 0.5436 | 0.5955 | 0.6862 | 3 | ✓ | ✓ | ✓ | 0.1894 | 0.3309 | 0.4581 |

Table 18: Results of architecture design ablation. Tested on 1000 randomly sampled architectures. Average over 3 trials. Results depict the Kendall Tau Correlation of FLAN with different DGF GAT module implementations.

| Samples | FLAN | $FLAN_{Arch2Vec}$ | $FLAN_{CATE}$ | $FLAN_{ZCP}$ | $FLAN_{CAZ}$ | Samples | FLAN | $FLAN_{Arch2Vec}$ | $FLAN_{CATE}$ | $FLAN_{ZCP}$ | $FLAN_{CAZ}$ |
|---|---|---|---|---|---|---|---|---|---|---|---|
| | NASBench-101 | | | | | | NASBench-201 | | | | |
| 8 | $0.3811_{0.0047}$ | $0.2748_{0.0348}$ | $0.4417_{0.0120}$ | $0.4590_{0.0033}$ | $0.4794_{0.0025}$ | 8 | $0.4101_{0.0434}$ | $0.4197_{0.0551}$ | $0.4276_{0.0512}$ | $0.4290_{0.0375}$ | $0.4299_{0.0495}$ |
| 16 | $0.5079_{0.0023}$ | $0.4458_{0.0009}$ | $0.4076_{0.0050}$ | $0.5501_{0.0012}$ | $0.4682_{0.0082}$ | 16 | $0.6124_{0.0025}$ | $0.6164_{0.0029}$ | $0.6108_{0.0023}$ | $0.6120_{0.0031}$ | $0.6188_{0.0036}$ |
| 32 | $0.5729_{0.0017}$ | $0.5121_{0.0028}$ | $0.5755_{0.0029}$ | $0.6024_{0.0023}$ | $0.5902_{0.0017}$ | 32 | $0.6917_{0.0009}$ | $0.6923_{0.0008}$ | $0.6912_{0.0010}$ | $0.6948_{0.0010}$ | $0.6888_{0.0011}$ |
| 64 | $0.6356_{0.0005}$ | $0.6067_{0.0014}$ | $0.6579_{0.0007}$ | $0.6945_{0.0008}$ | $0.6778_{0.0007}$ | 64 | $0.7608_{0.0006}$ | $0.7652_{0.0005}$ | $0.7693_{0.0006}$ | $0.7699_{0.0006}$ | $0.7573_{0.0003}$ |
| 128 | $0.7174_{0.0002}$ | $0.6910_{0.0003}$ | $0.7205_{0.0003}$ | $0.7586_{0.0001}$ | $0.7409_{0.0002}$ | 128 | $0.8280_{0.0001}$ | $0.8234_{0.0002}$ | $0.8204_{0.0002}$ | $0.8190_{0.0002}$ | $0.8274_{0.0003}$ |
| | DARTS | | | | | | $DARTS_{FixWD}$ | | | | |
| 8 | $0.0270_{0.0009}$ | $0.0464_{0.0013}$ | $0.0441_{0.0017}$ | $0.1589_{0.0034}$ | $0.1013_{0.0033}$ | 8 | $0.1006_{0.0097}$ | $0.1188_{0.0075}$ | $0.1196_{0.0077}$ | $0.1217_{0.0086}$ | $0.1130_{0.0074}$ |
| 16 | $0.0734_{0.0005}$ | $0.0903_{0.0015}$ | $0.0858_{0.0014}$ | $0.2890_{0.0112}$ | $0.2995_{0.0084}$ | 16 | $0.1724_{0.0104}$ | $0.1524_{0.0109}$ | $0.1698_{0.0070}$ | $0.1759_{0.0074}$ | $0.1674_{0.0078}$ |
| 32 | $0.1928_{0.0041}$ | $0.1643_{0.0014}$ | $0.1825_{0.0041}$ | $0.3767_{0.0033}$ | $0.3899_{0.0028}$ | 32 | $0.3214_{0.0047}$ | $0.2321_{0.0075}$ | $0.3203_{0.0044}$ | $0.3927_{0.0057}$ | $0.2562_{0.0033}$ |
| 64 | $0.3893_{0.0040}$ | $0.3390_{0.0075}$ | $0.4252_{0.0029}$ | $0.5468_{0.0003}$ | $0.5175_{0.0006}$ | 64 | $0.4750_{0.0011}$ | $0.3852_{0.0023}$ | $0.4830_{0.0018}$ | $0.4791_{0.0014}$ | $0.4535_{0.0035}$ |
| 128 | $0.4872_{0.0014}$ | $0.4006_{0.0050}$ | $0.5315_{0.0022}$ | $0.5845_{0.0005}$ | $0.5583_{0.0003}$ | 128 | $0.5546_{0.0008}$ | $0.4567_{0.0027}$ | $0.5567_{0.0004}$ | $0.5626_{0.0009}$ | $0.5449_{0.0012}$ |
| | ENAS | | | | | | $ENAS_{FixWD}$ | | | | |
| 8 | $0.0443_{0.0016}$ | $0.0431_{0.0017}$ | $0.0454_{0.0024}$ | $0.0939_{0.0028}$ | $0.0983_{0.0022}$ | 8 | $0.1655_{0.0149}$ | $0.1145_{0.0099}$ | $0.1406_{0.0119}$ | $0.1612_{0.0225}$ | $0.1763_{0.0244}$ |
| 16 | $0.1450_{0.0235}$ | $0.0663_{0.0026}$ | $0.1466_{0.0141}$ | $0.3350_{0.0122}$ | $0.2711_{0.0060}$ | 16 | $0.2672_{0.0147}$ | $0.2173_{0.0139}$ | $0.3103_{0.0093}$ | $0.3509_{0.0038}$ | $0.3357_{0.0067}$ |
| 32 | $0.2645_{0.0085}$ | $0.1389_{0.0068}$ | $0.2645_{0.0115}$ | $0.4348_{0.0039}$ | $0.3809_{0.0027}$ | 32 | $0.3871_{0.0087}$ | $0.2445_{0.0090}$ | $0.4034_{0.0065}$ | $0.4155_{0.0044}$ | $0.3927_{0.0036}$ |
| 64 | $0.3429_{0.0027}$ | $0.2910_{0.0016}$ | $0.3739_{0.0018}$ | $0.5145_{0.0020}$ | $0.4792_{0.0011}$ | 64 | $0.4704_{0.0027}$ | $0.3642_{0.0060}$ | $0.5025_{0.0002}$ | $0.5048_{0.0013}$ | $0.4610_{0.0022}$ |
| 128 | $0.4585_{0.0016}$ | $0.3852_{0.0013}$ | $0.4868_{0.0011}$ | $0.5683_{0.0003}$ | $0.5383_{0.0002}$ | 128 | $0.5288_{0.0004}$ | $0.4658_{0.0006}$ | $0.5428_{0.0002}$ | $0.5635_{0.0002}$ | $0.5335_{0.0009}$ |

Table 19: We study the effect of supplementing the FLAN with different representations.

| Samples | $FLAN^T$ | $FLAN^T_{Arch2Vec}$ | $FLAN^T_{CATE}$ | $FLAN^T_{ZCP}$ | $FLAN^T_{CAZ}$ | Samples | $FLAN^T$ | $FLAN^T_{Arch2Vec}$ | $FLAN^T_{CATE}$ | $FLAN^T_{ZCP}$ | $FLAN^T_{CAZ}$ |
|---|---|---|---|---|---|---|---|---|---|---|---|
| | NASBench-201 $\longrightarrow$ **NASBench-101** | | | | | | NASBench-101 $\longrightarrow$ **NASBench-201** | | | | |
| 2 | $0.5252_{0.0000}$ | $0.2981_{0.0285}$ | $0.5320_{0.0043}$ | $0.4702_{0.0008}$ | $0.4438_{0.0015}$ | 2 | $0.6437_{0.0006}$ | $0.4835_{0.0024}$ | $0.4885_{0.0015}$ | $0.3996_{0.0060}$ | $0.5902_{0.0005}$ |
| 4 | $0.5119_{0.0021}$ | $0.4066_{0.0071}$ | $0.5178_{0.0020}$ | $0.4594_{0.0026}$ | $0.4236_{0.0061}$ | 4 | $0.6208_{0.0007}$ | $0.5236_{0.0038}$ | $0.4902_{0.0036}$ | $0.4369_{0.0044}$ | $0.5293_{0.0130}$ |
| 8 | $0.5361_{0.0037}$ | $0.4717_{0.0040}$ | $0.5127_{0.0013}$ | $0.5171_{0.0016}$ | $0.4861_{0.0016}$ | 8 | $0.6474_{0.0072}$ | $0.5675_{0.0118}$ | $0.6062_{0.0022}$ | $0.5197_{0.0113}$ | $0.6059_{0.0078}$ |
| 16 | $0.5803_{0.0004}$ | $0.5236_{0.0054}$ | $0.5628_{0.0003}$ | $0.5422_{0.0027}$ | $0.5344_{0.0028}$ | 16 | $0.6923_{0.0014}$ | $0.6451_{0.0031}$ | $0.6491_{0.0033}$ | $0.6409_{0.0019}$ | $0.7162_{0.0031}$ |
| | ENAS $\longrightarrow$ **DARTS** | | | | | | $PNAS_{FixWD} \longrightarrow$ **$DARTS_{FixWD}$** | | | | |
| 2 | $0.4946_{0.0000}$ | $0.5141_{0.0000}$ | $0.5050_{0.0000}$ | $0.5811_{0.0000}$ | $0.5464_{0.0000}$ | 2 | $0.4186_{0.0000}$ | $0.2941_{0.0000}$ | $0.4151_{0.0000}$ | $0.4548_{0.0000}$ | $0.3557_{0.0000}$ |
| 4 | $0.5039_{0.0059}$ | $0.4402_{0.0087}$ | $0.4789_{0.0009}$ | $0.5433_{0.0067}$ | $0.5135_{0.0045}$ | 4 | $0.3944_{0.0004}$ | $0.3120_{0.0124}$ | $0.3791_{0.0013}$ | $0.4400_{0.0010}$ | $0.4194_{0.0016}$ |
| 8 | $0.5061_{0.0016}$ | $0.5210_{0.0022}$ | $0.5254_{0.0007}$ | $0.5852_{0.0016}$ | $0.5336_{0.0036}$ | 8 | $0.3722_{0.0013}$ | $0.3534_{0.0024}$ | $0.3929_{0.0039}$ | $0.4320_{0.0015}$ | $0.3859_{0.0030}$ |
| 16 | $0.5150_{0.0020}$ | $0.4836_{0.0025}$ | $0.4845_{0.0030}$ | $0.5940_{0.0014}$ | $0.5526_{0.0019}$ | 16 | $0.3653_{0.0013}$ | $0.3698_{0.0035}$ | $0.3898_{0.0032}$ | $0.4363_{0.0033}$ | $0.4167_{0.0008}$ |
| | DARTS $\longrightarrow$ **ENAS** | | | | | | NASNet $\longrightarrow$ **$ENAS_{FixWD}$** | | | | |
| 2 | $0.4526_{0.0000}$ | $0.4797_{0.0000}$ | $0.4442_{0.0000}$ | $0.5870_{0.0000}$ | $0.4391_{0.0000}$ | 2 | $0.3166_{0.0000}$ | $0.2474_{0.0000}$ | $0.4352_{0.0000}$ | $0.4566_{0.0000}$ | $0.4218_{0.0000}$ |
| 4 | $0.3980_{0.0044}$ | $0.3599_{0.0040}$ | $0.3916_{0.0075}$ | $0.4970_{0.0011}$ | $0.3743_{0.0078}$ | 4 | $0.2581_{0.0023}$ | $0.3001_{0.0227}$ | $0.3864_{0.0018}$ | $0.4471_{0.0018}$ | $0.3676_{0.0086}$ |
| 8 | $0.3908_{0.0064}$ | $0.4315_{0.0039}$ | $0.4094_{0.0024}$ | $0.5308_{0.0007}$ | $0.4184_{0.0017}$ | 8 | $0.2521_{0.0209}$ | $0.3984_{0.0047}$ | $0.3806_{0.0040}$ | $0.4602_{0.0011}$ | $0.4044_{0.0044}$ |
| 16 | $0.4323_{0.0012}$ | $0.4037_{0.0022}$ | $0.4337_{0.0012}$ | $0.5336_{0.0018}$ | $0.3688_{0.0087}$ | 16 | $0.3405_{0.0038}$ | $0.4126_{0.0024}$ | $0.3899_{0.0051}$ | $0.4361_{0.0028}$ | $0.4223_{0.0011}$ |

Table 20: Transferring predictors from one search-space to another (Source $\longrightarrow$ Target) benefits from supplementary encodings such as CATE and Arch2Vec.

| Forward | Backward | NB101 | NB201 | NB301 | Amoeba | PNAS | NASNet | $DARTS_{FixWD}$ | $ENAS_{FixWD}$ | TB101 |
|---|---|---|---|---|---|---|---|---|---|---|
| DGF | DGF | $0.7088_{0.0003}$ | $0.7981_{0.0004}$ | $0.7129_{0.0001}$ | $0.4200_{0.0003}$ | $0.3751_{0.0008}$ | $\mathbf{0.4191_{0.0038}}$ | $0.4632_{0.0003}$ | $0.4799_{0.0010}$ | $\mathbf{0.7939_{0.0001}}$ |
| GAT | GAT | $0.6535_{0.0000}$ | $0.7724_{0.0000}$ | $0.7938_{0.0001}$ | $0.3751_{0.0018}$ | $0.3570_{0.0037}$ | $0.3134_{0.0013}$ | $0.5441_{0.0005}$ | $0.4590_{0.0039}$ | $0.7458_{0.0002}$ |
| DGF+GAT | DGF | $0.7182_{0.0003}$ | $0.8106_{0.0000}$ | $0.8110_{0.0000}$ | $0.38577_{0.0024}$ | $0.3009_{0.0064}$ | $0.3173_{0.0043}$ | $0.5523_{0.0000}$ | $0.5257_{0.0021}$ | $0.7648_{0.0003}$ |
| **DGF+GAT** | **DGF+GAT** | $\mathbf{0.7322_{0.0002}}$ | $\mathbf{0.8200_{0.0004}}$ | $\mathbf{0.8202_{0.0000}}$ | $\mathbf{0.4594_{0.0000}}$ | $\mathbf{0.4225_{0.0004}}$ | $0.3870_{0.0099}$ | $\mathbf{0.5577_{0.0005}}$ | $\mathbf{0.5685_{0.0016}}$ | $0.7544_{0.0003}$ |

Table 21: We look at different GNN designs on a wider set of design spaces. 128 samples are used to train, and tested on the entire NAS space.

