# OpenReview forum: "Encodings for Prediction-based Neural Architecture Search"
_ICLR.cc/2024/Conference — Submitted to ICLR 2024_

### Official Review · Reviewer_14w6 · 2023-10-29

**Soundness:** 2 fair
**Presentation:** 1 poor
**Contribution:** 2 fair
**Rating:** 3
**Confidence:** 5

**Summary:**

This paper proposes FLAN, or FLow Attention for NAS, a neural predictor that incorporates a unified architecture encoding scheme to enable prediction across different NAS search spaces. The unified encoding is a mix of several unsupervised methods, like Arch2vec, CATE, and Zero-Cost Proxies, while the GNN predictor incorporates new mechanisms to potentially overcome the oversmoothing problem. Several design components of FLAN are ablated, and transfer tests are performed with an emphasis on sample efficiency.

**Strengths:**

Generalzing NAS predictors to cover multiple search spaces is essential and an important step forward.
For the most part, the paper is easy to read and follow early on. Figures 1-3 especially are nicely done.
There is detailed ablation on the design aspects of FLAN. Transfer experiments are performed, as is search.

**Weaknesses:**

The are issues with the contributions and statements made in this manuscript:

First, DGF: The author's point out that "GCNs are prone to an over-smoothing problem", although really this issue affects Graph Neural Networks (GNNs) in general. The author's then attempt to validate the efficacy of FLAN's predictor using the DGF and GAT in Table 1. I am not convinced by these experiments. GCN was proposed in 2016 and since then other GNN-types like GAT, GIN [1], GATv2 [2], etc., all of whose manuscripts demonstrate a superiority over GCN, so its no surprise that for FLOW, GAT > GCN. Also, it is unclear whether the DGF module used actually solves the oversmoothing mechanism at all. A simpler explanation is that adding skip-connections and more weights simply leads to better performance. Thus, results in Table 1 are inconsistent and do little to alleviate this concern, and it seems this paper cannot decide if it wants to be about correcting GNN problems or neural predictors in NAS.

Second, Unified Encodings for transferable NAS prediction. The author's cite Multi-Predict and GENNAPE. Given the focus on cell-based NAS, there is related work in this field called CDP [3] that is older than Multi-Predict/GENNAPE (both recent 2023 papers), but the authors seem unaware of. The form of encoding in this paper is weaker than related work for three reasons: First, there is no guarantee an architecture encoding will be unique as it possible for two different architectures to have almost the same ZCP scores/latency/FLOPs/etc. but be different structurally. 'Score-based' is a better descriptor for this encoding and I think the authors would be better off emphasizing that term to describe their method. Second, the use of "unique numerical indices" limits FLAN to a small number of predetermined search spaces so it cannot address the need to "validate NAS improvement without incurring the large compute cost of performing NAS on a new search space". Finally, the statement "However, search on a global graph structure—one that encompasses all NNs—can be an intractable search problem due to its size." is very weak. Large search spaces have never been an impediment even for early NAS methods [4][5], and it is still possible to perform search within cell-based NAS-Benchmarks even if using transferable encodings like CDP or GENNAPE.

Third, experimental inferences are not supported by the tables/figures. E.g., Fig. 4 t-SNE plots, specifically "In contrast, CATE doesn’t exhibit such distinct clusters, instead scattering NNs according to their computational complexity." Actually, the CATE plot shows a large number of individual clusters per search space, and the clusterings for each search space are close to each other. I am skeptical of making strong claims like that on a t-SNE plot as its very hard to interpret what each axis is measuring. Also, in Sec. 5, "Score-based encodings typically help prediction with low sample count but there are diminishing returns with more training samples". In Table 2 FLAN_{ZCP} sweeps performance for NB101 and ENAS, yet loses to TAGATES at the lowest data percent; also, the ZCP row itself does not boast impressive performance, but still monotonically increases with \%samples.

Results in Tables 3/4 do not instil confidence given the goal this paper is trying to solve. For Table 3, On NASBench-101, DARTS and ENAS (FixWD) FLAN+ZCP dominates all the time, less so on NASBench-201 and DARTS_{FixWD}, and for Table 4, it only dominates on ENAS<->DARTS. For a method that targets transferability when few samples in the Target search space are available, you would expect it to either dominate quite conclusively, or be able to provide reasoning why the best setting changing depending on the transfer target. That is, when the method is deployed in an actual scenario where you actually do not have a lot of labeled data (whereas here, you are just artificially limiting the amount), some inference can be made about which predictor configuration (+Arch2Vec, +CATE, +ZCP or +CAZ) is the best fit, given the Target dataset.
The lack of either of these mutes the usefulness of FLOW.

Next, NAS search results are not impressive. The author's deliberately limit themselves to cell-based benchmarks where there is a wealth of literature, but mostly on CIFAR-10. Unlike CDP they do not even evaluate on ImageNet, much less other tasks where the utility of NAS should be aimed.

Finally, the presentation in mixed. The introduction and related work are easy and nice to read, but beyond that, the writing is subpar, especially in Section 4.1 when describing the GNN/DGF. Floats from Table 1 onwards (DF+GAT x NB201 typo) leave a lot to be desired.
E.g., Figure 4 is a mess: Text is way too small and there are red dots which seemingly have no label in the legend. If the author's are that insistent on plotting out these many search spaces they should make use of different colors and marker shapes to help distinguish search spaces. Figures 5 and 6 use different marker shapes but its too small, some are the same color, and just generally difficult to make stuff out. Also, less is more - plots should have fewer entries, only several important variants so the viewer can actually see what is going on.

In summary, this paper starts off good with its writing, only to start making incorrect/unsupported statements/inferences in the methodology and results sections. This culminates in a very dense results section that fails to instill confidence in the efficacy of the proposed method should it be used down the line. For these reasons I recommend rejection of this manuscript.

References:

[1] Xu et al., "How Powerful are Graph Neural Networks?" In ICLR 2019.

[2] Brody, Alon and Yahav. "How Attentive are Graph Attention Networks?" In ICLR 2022.

[3] Liu et al., "Bridge the Gap Between Architecture Spaces via A Cross-Domain Predictor", in NeurIPS 2022.

[4] Zoph and Le. "Neural Architecture Search with Reinforcement Learning." In ICLR 2017.

[5] Bender et al., "Can Weight-Sharing Outperform Random Architecture Search? An Investigation with TuNAS." In CVPR 2020.

**Questions:**

- Page 6, just above section 4.2: "We train predictors on these search spaces by keeping separate DGF-GAT modules for the normal and reduce cells, and adding the aggregated outputs" - why this design choice? In graph frameworks like DGL and PyTorch-Geometric it is entirely possible to have single data samples (e.g., one prediction) that consist of multiple disconnected graphs.

- Section 5: "Contrary to prior work, we generate encodings for all architectures in the search space for evaluation." Then why does Table 2 list "Proporations of 7290 samples" for NAS-Bench-101?

- Not a question but I would suggest updating the bibliography entries as some influential works like NAS-Bench-301 were only on ArXiv for years but have now been formally published.

---

> ### Author Response · Authors · 2023-11-14
> **Response to reviewer 14w6 (Part 1)**
>
> We would like to thank you for your extremely thorough and insightful review. We agree with several of the concerns you have raised, and have done our best to address them. Specifically, we had made unsupported claims about the predictor design, that have now been fixed in our writing throughout the manuscript. We have also added further results for your consideration. Please find our full response below.
>
> `it is unclear whether the DGF module used actually solves the oversmoothing mechanism at all.`
>
> You are indeed right to point out that this issue affects GNNs in general, and that DGF does not necessarily solve the oversmoothing mechanism. We completely agree with this, and instead use empirical evidence to state that the skip connection in DGF may aid performance, and GAT ensemble seems to help. We believe there are 2 places where we make these claims that are not fully supported, we have now edited the text to remove these parts. Our paper’s key focus is on encodings for accuracy predictors within NAS, and our notes on predictor design were mainly done as a step to create a predictor to test out the performance of different encodings.
>
> `there is no guarantee an architecture encoding will be unique as it possible for two different architectures to have almost the same ZCP scores/latency/FLOPs/etc. but be different structurally. '`
>
> We completely agree. In fact, MultiPredict solely used ZCPs to encode architectures. These encodings do not provide uniqueness guarantees. This is one of the key reasons why, throughout our paper, we use these as supplemental encodings. These supplemental encodings are used in conjunction with the adjacency and operation matrices to build our accuracy predictors.
>
> `the use of "unique numerical indices" limits FLAN to a small number of predetermined search spaces `
>
> The training time for our predictor is very small (7.5 minutes). Thus, it is easy to simply _re-index_ the operation matrices and re-train a predictor to transfer to a novel search space. To avoid confusion, we have also added a short sentence discussing the cost of re-indexing a predictor such that it does not need to incur large compute costs on new search spaces. We also add a small section detailing this in the Appendix (A.4).
>
> `"However, search on a global graph structure—one that encompasses all NNs—can be an intractable search problem due to its size." is very weak. ` and `related work in this field called CDP [3] that is older than Multi-Predict/GENNAPE`
>
> We agree, thank you for bringing this to our attention. We have removed this weak claim and instead explicitly state our focus on cell-based NAS. We also acknowledge the work done in CDP for cross-domain prediction. We will try to compare to the CDP work–we are looking into it now.
>
> ` I am skeptical of making strong claims like that on a t-SNE plot as its very hard to interpret what each axis is measuring`
>
> Thank you for bringing this to our attention. We have shifted this to the appendix. Further, we have regenerated the T-SNE plot with fewer search spaces, and scaled the marker sizes to correlate with parameter counts. We have also added visualisation for ZCPs. Additionally, we removed any strong conclusions we made from this graph, and instead validated our finding further by adding comments on explicit correlation measurements in the Appendix.
>
> `FLAN_{ZCP} sweeps performance for NB101 and ENAS, yet loses to TAGATES at the lowest data percent`
>
> Upon request by reviewer nwAh, we repeated these experiments with 9 trials instead of 3 (Table 2,3,4). We have added results for NASBench301 as well. These two modifications were to be more directly comparable to TAGATES as requested by reviewer nwAH.
>
> Please note that out of the 12 tests comparing our performance with TAGATES, we only underperform by 0.0032 on the lowest data-percent on NASBench-201. In all other cases, FLAN ZCP outperforms TAGATES. We use this to show that we have a robust baseline predictor, and we offer further improvements by combining transfer learning and unified encodings.
>
> `you would expect it to either dominate quite conclusively…mutes the usefulness of FLOW.`
>
> We find that ZCP dominates most often, while sometimes CAZ may offer additional information by virtue of providing CATE and Arch2Vec encodings. Further, we justify/offer insights on the lukewarm result for NASBench-101 and NASBench-201 in the Appendix (Table 9) by empirically testing the kendall tau when using a single source space and multiple target spaces (Amoeba) and comparing those results with our current results in Table 4.

---

> > ### Author Response · Authors · 2023-11-14
> > **Response to reviewer 14w6 (Part 2)**
> >
> > `Figure 4 is a mess: Text is way too small and there are red dots which seemingly have no label in the legend. plots should have fewer entries, only several important variants so the viewer can actually see what is going on.`
> >
> > We completely agree with these comments. Initially, we aimed to maximize the number of results in the figures/graphs. We have taken this comment to heart, and instead shifted the full figures to the Appendix. We have regenerated Figure 4 and 5 with only a few important variants of our work. Further, we have made an effort to maximize the size of markers and text to improve the clarity of our figures. We have made appropriate changes to other sections as well to improve the quality of our presentation and we appreciate your comments regarding this.
> >
> > `DGF-GAT modules for the normal and reduce cells, and adding the aggregated outputs" - why this design choice? In graph frameworks like DGL and PyTorch-Geometric it is entirely possible`
> >
> > Our framework supports disconnected graphs in the same manner, and for few-shot learning, having a single shared module is beneficial, but when sufficient source-space samples are present, the ensembling helps.
> >
> > `"Contrary to prior work, we generate encodings for all architectures in the search space for evaluation." Then why does Table 2 list "Proporations of 7290 samples" for NAS-Bench-101?`
> >
> > We did this to compare fairly and directly to prior work. For example, TAGATES tests on 7290 samples instead of the entire search space for NB201. Table 2 is solely dedicated to comparing our predictor with existing work. For the rest of the paper, we report results on the entire search space where possible.
> >
> > In this work, we generate all encodings (individual and unified) for over 1.5 million neural networks, and 13 zero cost proxies for over 0.5 million neural networks and our code (to be released) allows experimentation with all of these encodings, and is no way limited to the 7290 architectures. This will hopefully enable future research on large NAS search spaces.
> >
> > In summary, we have slightly modified our text and T-SNE graph+interpretation to have a stronger focus on empirical evidence for the effectiveness of our methodologies, and re-ran key experiments in the tables with more trials upon request.
> >
> > We hope you have addressed your concerns, and would love to hear further feedback. Thank you!

---

> ### Author Response · Authors · 2023-11-20
> **Request for Further Feedback**
>
> We would like to thank you for your insightful and thorough assessment of our manuscript. We believe our revision addresses several of your key concerns. As we approach the final stages of the author-reviewer discussion phase, we request you to check our responses and paper revision and let us know if there are any additional comments or questions that we can address.
>
> Here is a quick summary of our response:
>
> -- Address concerns about oversmoothing claims; edited manuscript to clarify the role of skip connections and GAT-DGF ensemble and focused on encodings for accuracy predictors within NAS.
>
> -- Agree on limitations of ZCP/CATE/Arch2Vec not guaranteeing unique architecture encodings; our paper uses these as supplemental encodings alongside adjacency and operation matrices.
>
> -- Discussed re-indexing and re-training of FLAN for new search spaces to address limitations in operation matrices and search spaces; added details in Appendix A.4.
>
> --Revised weak claims and T-SNE interpretations, removing strong conclusions and focusing on empirical evidence; updated figures for clarity and re-ran key experiments with more trials for robust comparison with TAGATES and other methods.
>
> -- Improved presentation of key tables and figures.

---

> ### Author Response · Authors · 2023-11-21
> **Request for further feedback**
>
> As the author-reviewer discussion deadline is closing soon, we would like to request your reconsideration for our manuscript, following significant revisions based on reviewer feedback. We have made an effort to address several concerns raised, by refining our claims, enhancing the clarity of the paper and providing more robust empirical evidence. I am pleased to inform you that reviewer nwAh has updated their assessment to a positive acceptance, and reviewer oKjh has decided to retain their score after reviewing our updates. Your feedback is crucial in guiding the final version of our paper.

---

> > ### Comment · Reviewer_14w6 · 2023-11-22
> > **Maintaining Score**
> >
> > My apologies for the late response. I have read the rebuttal and combed through the revised manuscript.
> >
> > Overall, the rebuttal does not adequately address my concerns, so I will not be raising my
> > score. Below is a detailed summary:
> >
> > First, my original review challenged several of the claims made in the submission, such as oversmoothing in
> > GNNs motivating the use of DGF, transferable graph representations of GENNAPE/CDP and the t-SNE
> > plots. The response to these criticisms was to concede that their claims were false and comment them out
> > (DGF/GENNAPE/CDP) or rephrase what was said while moving it to the Appendix (t-SNE plots). In doing so the authors
> > undermine themselves, specifically their own motivations for the design decisions made in this submission compared to existing
> > literature as well as attempted insights which could be gleamed in a transferable NAS setting, which this paper is still lacking. Specifically,
> > - If the benefit of DGF is just empirically higher performance, are there not other, simpler mechanisms by which this
> >   can be achieved?
> > - Before, Table 1 is unchanged, despite what I said about GCN vs. GAT. If the baseline is patently easy to outperform (GAT got published by showings its better than GCN), it is not a good baseline.
> >
> > Second, the response to my criticism about 'Score-based' being a better descriptor for FLAN's architecture
> > representation was not convincing and half-hearted. The authors reply 'We have removed this weak claim and
> > instead explicitly state our focus on cell-based NAS', yet after checking the revised manuscript this change was
> > only made in one paragraph. The abstract, introduction, etc., still 'sell' the submission as quite broad rather than confined to cell-based benchmarks, so the authors are trying to 'have their cake and eat it too', so to speak. A similar rhetoric trick occurs when the author's respond to my question about the 7290 samples for NAS-Bench-101.
> >
> > Third, my concern regarding the unique numerical indices limiting FLAN was deflected, not addressed. The issue is not
> > how fast a neural predictor can be trained - the underlying motivation for neural predictors in the first place is a
> > quick avenue for performance evaluation, which entails quick training, and quick inference, so that reasoning fails.
> > Rather, since there are existing transferable encodings in the literature, why introduce a new one? If the
> > claim about GENNAPE's global graph structures leading to intractable search is false, why use
> > the unique numerical indices proposed with FLAN, which must be re-designed for new search spaces? Or, if the focus
> > must be cell-based, why not CDP's representation, which is already part of the peer-reviewed literature? Why not just
> > concat ZCP/CATE/Arch2Vec info to their existing architecture encodings? The rebuttal and revised manuscript evade
> > this concern.
> >
> > Fourth, revised plots either hide existing issues or reveal new ones. As previously said, the problematic
> > t-SNE figures from before, while regenerated/pruned, are now in the Appendix but the writing surrounding them just
> > rephrases what the authors said before: "CATE doesn't exhibit such distinct clusters, instead scattering NNs
> > according to their computational complexity" has been replaced with "This proximity is influenced by the
> > similarities in their parameter counts". Also, the cleaned-up Figure 4 reveals that FLAN has a
> > lot of instability, especially with the NAS-Bench-101 and NAS-Bench-201 plots. While other methods are not perfectly
> > monotonic, they do tend to be more consistent, meaning an end-user would be more confident believing that providing
> > more samples would yield more accurate predictions, even if they need to provide more samples upfront to begin with.
> >
> > Also, the revised manuscript has more trials for some experiments and performance decreases more than increases in Tables 2-5. Also, the major conclusion change, "FLAN to enhance its accuracy by up to 64%" now being reduced to 47%, is a huge
> > negative change.
> >
> > Other issues observed (some are just nitpicks):
> > - Typo in Table 1 has not been fixed.
> > - Not satisfied with the answer about dual encoders for search spaces with normal/reduction cells.
> > - Criticism about NAS search not being impressive is not addressed.
> > - Bibliography still hasn't been updated.
> > - Edits to tables were made to improve readability by removing the 4th significant digit, yet no rounding was done.
> > - Authors acknowledge CDP, state they will look into it, yet that has not been done besides 1 cite.

---

> ### Author Response · Authors · 2023-11-22
> **Request for further feedback**
>
> As the author-reviewer discussion deadline nears, we kindly request your reassessment of our manuscript. We have further made a final revision for the rebuttal period, where we re-format our tables to look more consistent in formatting, which we hope further improves the quality of presentation.

---

> ### Author Response · Authors · 2023-11-22
>
> We would like to thank the reviewer for their response before the rebuttal deadline. We understand several of your concerns, but we believe we have tried to address them previously, and we provide further elaboration below.
>
> `The response to these criticisms was to concede that their claims were false`
>
> The review process helps us refine our claims, and potentially identify unsupported statements. With your help, we have identified claims that are unsupported (not necessarily false), and adjusted them appropriately.
>
> `comment them out (DGF/GENNAPE/CDP) or rephrase what was said while moving it to the Appendix (t-SNE plots). In doing so the authors undermine themselves, `
>
> We benefit from the reviewer feedback on unsupported claims, and adjust our paper such that it better serves the NAS community. We move the plots to Appendix to make space for additional results requested by other reviewers. With your aid, we regenerated improved figures, and thus are able to have a more robust discussion in the appendix, and we add to the body of knowledge, and *not* to undermine ourselves. We also do not undermine our contributions, as we focus on **encodings and their role in cell based predictor-based NAS**, not necessarily designing the best graph neural network or addressing all forms of NAS.
>
> `If the benefit of DGF is just empirically higher performance, are there not other, simpler mechanisms by which this can be achieved?`
>
> Yes, we propose a new encoder that outperforms prior methods, we do not claim it is the best method possible in the field of graph neural network research. We empirically focus on **neural network encodings**.
>
>
>
>
> `(GAT got published by showings its better than GCN)`
>
> Again, we empirically focus on **neural network encodings**. The merit of adding GAT to the ensemble is not the sole contribution of this work.
>
> ` 'Score-based' being a better descriptor for FLAN's architecture representation was not convincing and half-hearted. `
>
> It is not a better descriptor, it is a supplementary encoding that was empirically shown to work well, and we use it as such.
>
> ` 'We have removed this weak claim and instead explicitly state our focus on cell-based NAS'`
>
> The claim was that global NAS is difficult, we only make this claim once, and thus, we removed it only once. This is part of our wider research question of enabling open-ended NAS but is not specifically addressed in this paper on NN encodings. Thus, we removed this sentence to keep the manuscript focused.
>
> `A similar rhetoric trick occurs when the author's respond to my question about the 7290 samples for NAS-Bench-101.`
>
> This is **not true**, the other tables present results on ALL NASBench-101 architectures. 7290 is explicitly to compare with TA-GATES and we still report full results.
>
> `If the claim about GENNAPE's global graph structures leading to intractable search is false, why use the unique numerical indices proposed with FLAN, which must be re-designed for new search spaces?`
>
> This is false, it does not need to be re-designed, re-indexing is trivial (counting and converting to one-hot) and search-space agnostic. We can indeed use ZCP or other metrics, but indexing is our selected method differentiating between operations in search space, and it works well as shown by our experiments. We definitely believe that this warrants further investigation in NAS.
>
> `Why not just concat ZCP/CATE/Arch2Vec info to their existing architecture encodings?`
>
> We explicitly do this, it is “CAZ”. However, as you mentioned (and we acknowledge), these representations are not unique, whereas indexes are unique by construction.
>
> `instead scattering NNs according to their computational complexity" has been replaced with "This proximity is influenced by the similarities in their parameter counts".`
>
> Computational complexity == FLOPs/Parameter Count, as described in the CATE [1] paper.
>
> `FLAN has a lot of instability, especially with the NAS-Bench-101 and NAS-Bench-201 plots.`
>
> This is explained in the Appendix, along with results for Amoeba.
>
> `"FLAN to enhance its accuracy by up to 64%" now being reduced to 47%, is a huge negative change.`
>
> We do not believe it is a huge negative change, as it still shows considerable improvement relative to prior work. This revision is the result of running more seeds as requested by other reviewers.
>
> We thank the reviewer for their detailed and constructive feedback. We hope you will reconsider your score, keeping in mind that our work primarily focuses on the **evaluation of neural architecture encodings in NAS**. We believe that many of your remaining concerns are not regarding our core contribution on studying DNN encodings, rather the predictor design (created as a vehicle to study encodings) and some (valid and helpful) nitpicks on result presentation.
>
> [1] Yan, S., Song, K., Liu, F., & Zhang, M. (2021). CATE: Computation-aware Neural Architecture Encoding with Transformers.

---

### Official Review · Reviewer_nwAh · 2023-11-01

**Soundness:** 3 good
**Presentation:** 3 good
**Contribution:** 3 good
**Rating:** 6
**Confidence:** 4

**Summary:**

This paper introduces a hybrid architecture performance predictor (FLAN) based on GCNs which improves sample efficiency. FLAN combines DGF with GAT, and further introduces a backward graph flow. It includes learnable operation embeddings and allows for concatenation of additional encodings, including score-based and unsupervised learned encodings. FLAN further enables transfer learning across cell-based search spaces, by appending unique numerical indices to cell encodings of each space.

The paper demonstrates both efficacy and sample efficiency in performance prediction, as well as sample efficiency in sample-based neural architecture search (NAS). Furthermore, the authors conduct transfer experiments, revealing the transferability between certain search spaces.

**Strengths:**

The paper is well-written and demonstrates an integration of ideas from prior work, further enhancing them to achieve SOTA sample efficiency in performance prediction and in sample-based NAS. Additionally, it shows superior Kendal tau correlation in performance prediction. The method also permits integration of new encodings, and hence allows to take advantage of developments in architecture encoding, such as new ZCPs. Furthermore, the unified encoding facilitates transfer learning across different cell-based search spaces.

**Weaknesses:**

- The Authors state that improvements gained in sample efficiency by pre-training FLAN on a source search space do not include the cost to pre-train the model. However, this makes sense only if the pre-training is done on a single source space and then transferred to any other space. From table 4, this doesn’t seem to be the case, and each target space has its own source space.

- Experiments on sample-based NAS are limited to NB101. Extending this (for example to NB201) would give a better assessment of the performance of FLAN.

- The applicability of the method is limited to cell-based spaces.

**Questions:**

Clarification questions and comments:

- To make a better comparison with TA-GATES, NB301 would be a natural choice to include in table 2.

- In Figs.5,6 the source search spaces used for pre-training are not clearly specified. Are these the same as Table.4?

- In Table.15, for Amoeba, NB301 and TB101, what are the source search spaces used for pre-training FLAN^T?

- Is Table.15 supposed to generalize Tables.3,4 (or Tables.17,18)? If so, why don’t the numbers match? See e.g. FLAN_ZCP for 128 samples on NB101.

- In the experiments of Table.2 (or Table.16) the number of trials for FLAN seems to be 3. Is this consistent with the number of trials used for TA-GATES reported in the tables?

- In the line above Table.4, “Over baseline FLAN, incorporating ZCP encoding improves predictor accuracy by 64% on average”, how can we see this from the tables?

- In Table.4, is the entire source space used for pre-training?

- In the NAS paragraph of page 9: “Our FLAN^T_{CAZ} is able to improve the sample efficiency of end-to-end NAS on NB101 by 2.12× over other methods”, shouldn’t this be FLAN_{CAZ}?

- In Table.5, the search method Zero-Cost NAS (W) is not described and is not cited (is this Zero-Cost Warmup?).

- In Table.15, the space DARTS_{LRWD} is not described.

---

> ### Author Response · Authors · 2023-11-14
> **Response to reviewer nwAh**
>
> We would like to thank the reviewer for their constructive review of our work, and appreciate the valuable suggestions on further experiments to validate and justify our results. We have taken your concerns fully into consideration and added more results to address them in the paper and in the Appendix (highlighted in red in the revised manuscript). Below, we summarize the main changes and respond to your questions.
>
> `From table 4, this doesn’t seem to be the case, and each target space has its own source space`
>
> In the main paper, we hope to demonstrate that our framework supports transfer from an arbitrary source space to a target space. We believe that demonstrating this ability is key to universal sample efficient NAS. However, we also acknowledge the importance of demonstrating the transfer of a predictor from a single search space to multiple target spaces. With this in mind, we have added a table to the Appendix (Table 9), where we take Amoeba as our choice of source space, and transfer the trained predictor to NASBench-101, NASBench-201, DARTS, DARTS Fix W-D, ENAS, ENAS Fix W-D. Our conclusion further emphasises the trend which favours ZCP and CAZ supplementary encodings during transfer. We hope this helps in evaluating the effectiveness of predictor transfer.
>
> `Experiments on sample-based NAS are limited to NB101. Extending this (for example to NB201) would give a better assessment of the performance of FLAN.`
>
> We completely agree. We evaluate our NAS method on three search spaces in the main paper (Figure 5) and on all search spaces in the appendix (Figure 10). Please note that Table 6 is formatted in that manner to be consistent with prior literature (e.g. BRP-NAS). which report results on this NAS algorithm variant only on NASBench-101, and contextualise our NAS algorithm with respect to other relevant work.
>
> `To make a better comparison with TA-GATES, NB301 would be a natural choice to include in table 2.`
>
> We agree, one of the reasons why we did not include NB301 is because we were not able to trace back the networks used for training TA-GATES on NB301 to the original NB301 library. We have now added the NB301 results in Table 2, and FLAN ZCP performs consistently very well in our tests.
>
> `In Figs.5,6 the source search spaces ... specified. ...same as Table.4?`
>
> Thank you for bringing this to our attention. Yes, it is indeed the same as Table 4. We have added a clarification to the caption to explicitly state this.
>
>
> `In Table.15, for Amoeba, NB301 and TB101, what are the source search spaces used for pre-training FLAN^T?`
>
> Thank you for bringing this to our attention, we have added another table that shows the source to target search space mapping. These experiments will also be in ready-to-go scripts in our code for further clarity.
>
> `In the experiments of Table.2 (or Table.16) the ... of trials used for TA-GATES reported in the tables?`
>
> Thank you for bringing this to our attention. To fairly evaluate our work with TA-GATES, we have now re-evaluated our main tables for 9 trials. All our conclusions remain the same despite minor numerical differences.
>
> `Is Table.15 supposed to generalize ... See e.g. FLAN_ZCP for 128 samples on NB101.`
>
> There is a slight numerical mismatch because the larger table has fewer trials. As mentioned above, to match the trials with TAGATES, we updated our main paper results to be reporting on 9 trials.
>
> `predictor accuracy by 64% on average”, how can we see this from the tables?`
>
> These calculations are made as shown below:
> ```
> >>> flanbase = [0.4424,0.3898,...,0.4876,0.5252]
> >>> flanzcp = [0.4996,0.5655,...0.5231,0.5511]
> >>> sum([100*(flanzcp[idx] - flanbase[idx])/flanbase[idx] for idx in range(len(flanzcp))])/len(flanzcp)
> >>> 64.47
> ```
> The numbers in the flanbase and flanzcp list represent the kendall tau correlation for tests in the table (full list excluded due to character limit but can be provided upon request), and we calculate the average of the percentage improvement in base v.s. ZCP supplementary encoding.
>
> ` is the entire source space used for pre-training?`
>
> No, only 512 samples are used for pre-training on the source space. We detail this in the Appendix, but will also add this to the main paper for clarity. Thank you for pointing this out.
>
> ` “Our FLAN^T_{CAZ} is able to ... by 2.12× .... shouldn’t this be FLAN_{CAZ}?`
>
> Thank you for the correction, you are indeed right. We have made the appropriate changes.
>
> ` the search method Zero-Cost NAS (W) ... DARTS_{LRWD} is not described.`
>
> We have added a citation for this work. We have added the description in the appendix, thank you for bringing this to our attention.
>
> In summary, we have added some cross-dataset and task results (Table 5), provided further context and clarification based on the suggestions and repeated experiments with more trials, which further support our findings.. Please let us know if we have address all of your comments, or if you have any further questions/concerns. Thank you.

---

> ### Author Response · Authors · 2023-11-20
> **Request for Further Feedback**
>
> We would like to thank you for your insightful and thorough assessment of our manuscript. We believe our revision addresses several of your key concerns. As we approach the final stages of the author-reviewer discussion phase, we request you to check our responses and paper revision and let us know if there are any additional comments or questions that we can address.
>
> Here is a quick summary of our response:
>
> -- Expanded the scope of transfer experiments to include Amoeba as a source space transferring to multiple target spaces, as detailed in the new Table 9 in the Appendix, demonstrating the effectiveness of predictor transfer across various search spaces.
>
> -- Addressed the need for broader NAS evaluations by adding NASBench-301 results in Table 2, showing consistent performance of FLAN ZCP across different benchmarks. Also informed reviewer of our NAS evaluation in Figure 10.
>
> -- Clarified the source search spaces used in Figures 5 and 6, and added a new table outlining source to target search space mapping for further clarity.
>
> --Updated experimental trials to match those of TA-GATES for a fair comparison, and clarified numerical discrepancies and methodological details in response to specific queries about Tables 2, 15, and other results.

---

> > ### Comment · Reviewer_nwAh · 2023-11-20
> >
> > I would like to thank the Authors for their response. Just a couple of clarification questions:
> > - Regarding the statement “There is a slight numerical mismatch because the larger table has fewer trials”, do I understand correctly that in the previous manuscript where the mismatch was present all results were reported for 3 trials?
> > - Regarding the statement “We evaluate our NAS method on three search spaces in the main paper (Figure 5) and on all search spaces in the appendix (Figure 10)”, Figures 5 and 10 evaluate different versions of FLAN but comparison to prior work is still limited to NB101 in Table.6. Are there any limitations in comparing FLAN with prior sample-based NAS methods for example on NB201 as in BRP-NAS Fig.6?

---

> > > ### Author Response · Authors · 2023-11-20
> > > **Clarification on trials and more NAS results.**
> > >
> > > Thank you for your response. We appreciate the opportunity to provide further clarification on the points raised.
> > >
> > > In the initial revision of our paper, the primary results (as seen in Table 3) were based on 7 trials, detailed in Line 100 of our code repository (https://anonymous.4open.science/r/flan_nas-433F/correlation_trainer/large_run_slurms/unified_joblist.log). The results presented in the large table were based on 3 trials (Line 255 in the same code repository). We acknowledge this inconsistency and have updated the main tables with 9 trials. We will update the hyper-parameter table (Table 8) appropriately, and re-generate the large table in the Appendix with 9 trials. The conclusions from our results do not change, and are consistent across trials. We will also open-source our code.
> > >
> > > There are no inherent limitations in comparing FLAN with other sample-based NAS methods, such as NB201, as demonstrated in BRP-NAS. Kindly note that Figure 6 of BRP-NAS demonstrates NAS on NASBench-201 CIFAR100. **To further corroborate the effectiveness of our predictor in NAS settings, we add support for NASBench-201 CIFAR100 in our code-base, and generate search results to compare with BRP-NAS, Aging Evolution, REINFORCE and Random Search. We add this to the Appendix (Figure 7).** It can also be accessed at (https://ibb.co/rsbjG4g). Because of significant variances in NAS algorithms in existing literature, comparison to prior work in Table 6 is indeed limited to NB101, following the setting of a key paper in low-cost NAS [1].  Our research primarily focuses on the efficacy of encodings in training neural accuracy predictors. While exploring sample-based NAS methods is valuable, our study's scope is concentrated on sample-efficient accuracy predictor training.
> > >
> > > We hope these clarifications adequately address the concerns raised in your review. We request you to reassess your rating of our paper in light of these updates, and are eager to provide further clarifications as required.
> > >
> > > [1] Abdelfattah, M. S., Mehrotra, A., Dudziak, Ł., & Lane, N. D. (2021). Zero-Cost Proxies for Lightweight NAS.

---

> > > > ### Comment · Reviewer_nwAh · 2023-11-20
> > > >
> > > > I thank the Authors for their response and appreciate their efforts. I will raise my rating to 6.
> > > >
> > > > (There’s a minor typo in the caption of Fig.7 mentioning “CIFAR10” as the dataset)

---

> > > > > ### Author Response · Authors · 2023-11-20
> > > > >
> > > > > We have submitted a revision of the paper where we fix the typo in Figure 7. We would like to thank you for your insightful and thorough review, and appreciate your reassessment.

---

> > > > > > ### Author Response · Authors · 2023-11-21
> > > > > >
> > > > > > Hello, we just wanted to send a quick reminder as the deadline for our discussion period is approaching. If you have a moment, could you please update your review score? Thank you so much for your time and your valuable feedback!

---

### Official Review · Reviewer_oKjh · 2023-11-10

**Soundness:** 4 excellent
**Presentation:** 4 excellent
**Contribution:** 4 excellent
**Rating:** 8
**Confidence:** 5

**Summary:**

This paper conducts a comprehensive study of encodings for NAS, as well as introduces a new NAS performance predictor that utilizes a GNN-based encoding which can be supplemented with other encodings such as zero-cost proxies. The new encoding is unified across search spaces, which enables transferrability of the pretained predictor to new spaces and more sample efficiency when finetuning it compared to training from scratch. The authors evaluate their predictor on a plethora of search spaces and release a dataset containing all their hardcoded and learned encodings for 1.5 million architectures across all their evaluated spaces.

**Strengths:**

- The motivation to have an unified encoding across NAS spaces is important and as the authors mention, this is relevant when it comes to transfer learning across spaces and tasks.

- The authors propose a new hybrid encoder that outperforms prior encodings and allows transferrability  of predictors to new search spaces. This leads to improved sample efficiency compared to training predictors from scratch on a new search space.

- Large-scale study of NAS encodings over 13 NAS search spaces with 1.5 million architectures in total across different tasks and datasets.

- The authors provide the code necessary to run their methods and reproduce their experiments. Moreover, the authors also release the dataset containing the encodings (both hard-coded and learned), which is very useful for future research on the NAS field.

- The paper is really well-written and easy to follow. I enjoyed reading it.

**Weaknesses:**

- It seems that the performance predictor is transferable across search spaces, and can relatively predict the ranking good. However, as far as I saw this is done only on CIFAR-10, right? That means that if one wants to transfer a learned predictor on a new dataset, that would not be feasible with FLAN, or otherwise one would need to train FLAN on the said dataset from scratch.

- Other than this, I do not have any major weaknesses regarding this paper. I think that this is an important work for the NAS community.

**Minor**

- it would be great to refer to [1] and [2] that compares performance predictors for NAS on a variety of benchmarks. In [2] the authors also study the transferrability across spaces, though somehow orthogonal to this work since that transferrability is for predictors' traning hyperparameters.

**References**

[1] https://arxiv.org/abs/2104.01177

[2] https://arxiv.org/abs/2201.13396

**Questions:**

- Can the authors provide some experiments where they demonstrate the runtime of their performance predictor?

- In the unified encodings, the authors concatenate a unique search space index to every operation. As far as I understand this requires to know apriori the number and type of search spaces that you are training FLAN with. Maybe I understood this wrong, but would the choice of operations index for a search space impact the predictions when transferring to a new search space? For example, if search space A and B are similar, while search space A and C more distinct, would using index say 1, 2 and 10 for A, B and C, respectively, be a more reasonable choice than using 1, 10 and 2? Or is this invariant to the index choice?

- Did authors evaluate their simple NAS method in section 5 (Table 5) on other search spaces except NAS-Bench-101?

- In Table 4 it seems that the performance of FLAN + ZCP, almost in all settings, especially ENAS -> DARTS and vice verca, deteriorates as you increase the number of samples. Do the authors have any intuition why this is the case? Did they try with more number of samples (say up to 50 as in Table 5?

---

> ### Author Response · Authors · 2023-11-14
> **Response to reviewer oKjh**
>
> Thank you for your thoughtful and constructive review of our work. We appreciate your positive comments on the scale and significance of our study. We have taken your comments into consideration and added several results that further support our conclusions in the paper (highlighted in red in the updated manuscript).. Below, we respond to your questions and summarize our changes to the paper..
>
> `this is done only on CIFAR-10, right?`
>
> In this paper, we generate encodings for a large set of search spaces, one of them being TransNASBench-101 Micro, which covers several vision tasks. Additionally, our NDS results can be extended to ImageNet as well. In an effort to address this weakness, we have added two tables in the main paper (Table 5) which demonstrates the effectiveness of transferring predictors across tasks _within_ TransNASBench-101 Micro, as well as across datasets (NDS CIFAR-10 to ImageNet).
>
> `Demonstrate the runtime of performance predictor`
>
> We believe that runtime performance is very important when prototyping search spaces for accuracy. Our training time for a transfer experiment is 7.5 minutes, and the transfer time (to adapt to a new domain) is less than 1 minute. Further, for inference, we are able to evaluate an average of 160 neural architectures per second.  We have added this to the appendix (A.4).
>
> `deteriorates as you increase the number of samples.`
>
> We test this in more depth, specifically on the task of NDS CIFAR-10 to NDS ImageNet transfer, and find that in Table 5, more samples generally improves performance (On NDS DARTS, ENAS, NASNet and PNAS). There is some ambiguity when there are fewer than 8 samples (As seen on TB101), and this is because for such low sample-counts, the predictor performance variance could also be largely dictated by the specific NN architectures used for adaptation, and not necessarily just the learning algorithm. Future work could look at finding methods to sample better NN architectures for few-shot learning.
>
> `the authors concatenate a unique search space index to every operation. As far as I understand this requires to know apriori the number and type of search spaces`
>
> This observation is correct. However, we do not think that it is a major concern due to the efficiency of predictor training. One can simply _re-index_ the search spaces and re-train the FLAN model in under 10 minutes on a single consumer GPU. Generating the Arch2Vec and CATE representations will take 20 and 40 minutes respectively. We believe that due to the simplicity and effectiveness of this approach, it is very reasonable to adapt the predictor to more search spaces. We have further added comments on this in the paper (Section 4.3)
>
> `Did authors evaluate their simple NAS method in section 5 (Table 5) on other search spaces except NAS-Bench-101?`
>
> Yes, we evaluate our NAS method on three search spaces in the main paper (Figure 5), and on all search spaces in the appendix (Figure 9 and 10). Please note that Table 6 is formatted in such a way to be consistent and directly comparable with prior literature (that also used the same format) in a clear manner.
>
> In summary, we have made several improvements in the writing and graphs of the paper. Further, we have generated transfer results on TransNASBench-101 and NDS Imagenet, as well as benchmarked runtime performance in the Appendix. Upon request by Reviewer nwAh, we have updated the primary results in paper to average over 9 trials instead of 3. This is done to decrease variance and to be directly comparable to prior work (TAGATES) which also used 9 trials. All our conclusions stand unchanged. We believe this paper is of significant value to the NAS community, as we generate and catalogue encodings for over 1.5 million neural networks, and build a state-of-the-art predictor. We then take the predictor and combine encodings to improve sample efficiency of accuracy predictors by transfer learning. We demonstrate this effectiveness across search spaces, data-sets (CIFAR-10 to Imagenet) as well as computer vision domains (TransNASBench-101 Micro tasks). We also demonstrate NAS on several search spaces. We will open source our code to support future research in this field. We hope you have addressed your concerns, and would love to hear further feedback. Thank you!

---

> > ### Comment · Reviewer_oKjh · 2023-11-16
> > **Keeping my score**
> >
> > Thank you for your response. After reading the other reviews and the authors' response to them I agree with most of their concerns. However, I still think that this is a valuable work to the NAS community that provides a lot of insights by empirically evaluating the predictors across multiple search spaces and investigating their transferrability properties. Therefore, I decide to keep my score and recommend acceptance.

---

> > > ### Author Response · Authors · 2023-11-20
> > >
> > > Thank you for retaining your score and recommending acceptance.

---

### Meta-Review · Area_Chair_t8BV · 2023-12-09

**Metareview:**

This paper proposes a study of encodings used in NAS and proposes a performance predictor (FLAN) using a GNN-based encoding. Further encodings such as ZCP can be used in combination. The main merit of the new, GNN-based encoding is that is can be used across search spaces, so that pretrained predictors can be transferred from one search space to another to allow for fine-tuning.
The paper has received scores between 3 and 8, rating the strengths and weaknesses of the paper differently.
Yet, they agree on the strengths and weaknesses of the paper (the merit of a transferable encoding with high performance prediction ability  versus the lack of discussion of the impact of individual components (GAT vs GCN for example) and the missing integration of recent prior work).
The most important  weaknesses that are not addressed in the rebuttal are in the lack of comparison of the proposed approach to the previous papers

Liu et al., "Bridge the Gap Between Architecture Spaces via Cross-Domain Predictor", in NeurIPS 2022.
Mills et al., "GENNAPE: Towards Generalized Neural Architecture Performance Estimators", in AAAI-23.

The authors are encouraged to add GENNAPE to their framework for an improved future submission.

**Justification For Why Not Higher Score:**

The proposed methods shows clear merits, however, currently fails to integrate in the context of recent prior work.

**Justification For Why Not Lower Score:**

N/A

---

### Decision · Program_Chairs · 2024-01-16

Reject